# SAGDA: Achieving $\mathcal{O}(\epsilon^{-2})$ Communication Complexity in Federated Min-Max Learning

**Haibo Yang**
Dept. of ECE
The Ohio State University
Columbus, OH 43210
`yang.5952@osu.edu`

**Zhuqing Liu**
Dept. of ECE
The Ohio State University
Columbus, OH 43210
`liu.9384@osu.edu`

**Xin Zhang**
Dept. of Statistics
Iowa State University
Ames, IA 50010
`xinzhang@iastate.edu`

**Jia Liu**
Dept. of ECE
The Ohio State University
Columbus, OH 43210
`liu@ece.osu.edu`

## Abstract

Federated min-max learning has received increasing attention in recent years thanks to its wide range of applications in various learning paradigms. Similar to the conventional federated learning for empirical risk minimization problems, communication complexity also emerges as one of the most critical concerns that affects the future prospect of federated min-max learning. To lower the communication complexity of federated min-max learning, a natural approach is to utilize the idea of infrequent communications (through multiple local updates) same as in conventional federated learning. However, due to the more complicated inter-outer problem structure in federated min-max learning, theoretical understandings of communication complexity for federated min-max learning with infrequent communications remain very limited in the literature. This is particularly true for settings with non-i.i.d. datasets and partial client participation. To address this challenge, in this paper, we propose a new algorithmic framework called stochastic sampling averaging gradient descent ascent (SAGDA), which i) assembles stochastic gradient estimators from randomly sampled clients as control variates and ii) leverages two learning rates on both server and client sides. We show that SAGDA achieves a linear speedup in terms of both the number of clients and local update steps, which yields an $\mathcal{O}(\epsilon^{-2})$ communication complexity that is orders of magnitude lower than the state of the art. Interestingly, by noting that the standard federated stochastic gradient descent ascent (FSGDA) is in fact a control-variate-free special version of SAGDA, we immediately arrive at an $\mathcal{O}(\epsilon^{-2})$ communication complexity result for FSGDA. Therefore, through the lens of SAGDA, we also advance the current understanding on communication complexity of the standard FSGDA method for federated min-max learning.

## 1 Introduction

Recently, min-max optimization has drawn considerable attention from the machine learning community. Compared with conventional minimization problems (e.g., empirical risk minimization), min-max optimization has a richer mathematical structure, thus being able to model more sophisticated learning problems that emerge from ever-emerging applications. In particular, the subclass

36th Conference on Neural Information Processing Systems (NeurIPS 2022).

of nonconvex-concave and nonconvex-PL (Polyak-Łojasiewicz) min-max problems has important applications in, e.g., AUC (area under the ROC curve) maximization [1, 2], adversarial and robust learning [3, 4], and generative adversarial network (GAN) [5]. The versatility of min-max optimization thus sparks intense research on developing efficient min-max algorithms. In the literature, the family of primal-dual stochastic gradient methods is one of the most popular and efficient approaches. For example, the stochastic gradient descent ascent (SGDA) method in this family has been shown effective in centralized (single-machine) learning, both theoretically and empirically. However, as over-parameterized models (e.g., deep neural networks) being more and more prevalent, learning on a single machine becomes increasingly inefficient. To address challenge, large-scale distributed learning emerges as an effective mechanism to accelerate training and has achieved astonishing successes in recent years. Moreover, as more stringent data privacy requirements arise in recent years, centralized learning becomes increasingly infeasible due to the prohibition of data collection. This also motivates the need for distributed learning without sharing raw data. Consequently, there is a growing need for distributed/federated min-max optimization, such as federated deep AUC maximization [6, 7], federated adversarial training [8] and distributed/federated GAN [9–11].

Similar to conventional federated learning for minimization problems, federated min-max learning enjoys benefits of parallelism and privacy, but suffers from high communication costs. One effective approach to reduce communication costs is to utilize *infrequent* communications. For example, in conventional federated learning for minimization problems, the FedAvg algorithm [12] allows each client performs multiple stochastic gradient descent (SGD) steps to update the local model between two successive communication rounds. Then, local models are sent to and averaged periodically at the server through communications. Although infrequent communication may introduce extra noises due to *data heterogeneity*, FedAvg can still achieve the same convergence rate as distributed SGD, while having a significant lower communication complexity. Inspired by the theoretical and empirical success of FedAvg, a natural idea to lower the communication costs of federated min-max optimization is to utilize infrequent communication in the federated version of SGDA. Despite the simplicity of this idea, existing works can only show unsatisfactory convergence rates ($\mathcal{O}(1/\sqrt{mT})$ [13] and $\mathcal{O}(1/(mKT)^{1/3})$ [14]) for solving non-convex-strongly-concave or non-convex-PL by federated SGDA with infrequent communication ($m$ is the number of clients, $K$ is the number of local steps, and T is the number of communication rounds). These convergence rates do not match with that of the FedAvg method. These unsatisfactory results are due to the fact that federated min-max optimization not only needs to address the same challenges in conventional federated learning (e.g., data heterogeneity and partial client participation), but also handle the more complicated inter-outer problem structure. Thus, a fundamental question in federated min-max optimization is: *Can a federated SGDA-type method with infrequent communication provably achieve the same convergence rate and even the highly desirable linear speedup effect for federated min-max problems?*

In this paper, we answer this question affirmatively. The main contributions of this paper are summarized as follows:

- We propose a new algorithmic framework called SAGDA(stochastic sampling averaging gradient descent ascent), which assembles stochastic gradient estimators as control variates and leverages two learning rates on both server and client sides. With these techniques, SAGDA *relaxes* the restricted "bounded gradient dissimilarity" assumption, while still achieving the same convergence rate with low communication complexity. We show that SAGDA achieves the highly desirable linear speedup in terms of both the number of clients (even with partial client participation) and local update steps, which yields an $\mathcal{O}(\epsilon^{-2})$ communication complexity that is orders of magnitude lower than the state of the art in the literature of federated min-max optimization.

- Interestingly, by noting that the standard federated stochastic gradient descent ascent (FSGDA) is in fact a "control-variant-free" special version of our SAGDA algorithm, we can conclude from our theoretical analysis of SAGDA that FSGDA achieves an $\mathcal{O}(1/\sqrt{mKT})$ convergence rate for non-convex-PL problems with full client participation, which further implies the highly desirable linear speedup effect. This improves the state-of-the-art result of FSGDA by a factor of $\mathcal{O}(1/(mKT)^{1/6})$ [14] and matches the optimal convergence rate of non-convex FL. Therefore, through the lens of SAGDA, we also advance the current understanding on the communication complexity of the standard FSGDA method for federated min-max learning.

Table 1: Number of communication rounds and stochastic gradients per client to reach $\epsilon$-stationary point ($\|\nabla\Phi\| \leq \epsilon$) for federated non-convex-PL min-max learning, denoted as communication and client sample complexity. We omit the higher orders. Here $m$ is the number of clients. BGD means bounded gradient dissimilarity, which requires bounded data heterogeneity. SAGDA supports client sampling and does not require BGD assumption.

| Methods | BGD Assumption | Client Sampling? | Per-Client Sample Complexity | Communication Complexity |
|---|---|---|---|---|
| SGDA | – | – | $\epsilon^{-4}$ | – |
| Local SGDA [14] | ✔ | ✗ | $\max\{\epsilon^{-4}, \frac{1}{m^{2/3}}\epsilon^{-6}\}$ | $\mathcal{O}((1/m)\epsilon^{-6})$ |
| (Momentum) Local SGDA [15] | ✔ | ✗ | $\mathcal{O}((1/m)\epsilon^{-4})$ | $\mathcal{O}(\epsilon^{-3})$ |
| CD-MAGE [13] | ✔ | ✔ | $\mathcal{O}((1/m)\epsilon^{-4})$ | $\mathcal{O}((1/m)\epsilon^{-4})$ |
| SAGDA (Cor. 1) | Not needed | ✔ | $\mathcal{O}((1/m)\epsilon^{-4})$ | $\mathcal{O}(\epsilon^{-2})$ |
| FSGDA (Cor. 2 3) | ✔ | ✔ | $\mathcal{O}((1/m)\epsilon^{-4})$ | $\mathcal{O}(\epsilon^{-2})$ |

The rest of the paper is organized as follows. In Section 2, we review related work. In Section 3, we first introduce SAGDA And its convergence analysis, and then build the connection between SAGDA and FSGDA. We present numerical results in Section 4 and conclude the work in Section 5. Due to space limitation, we relegate all proofs and some experiments to the supplementary material.

## 2   Related work

**1) Federated Learning:** In federated learning (FL), the seminal federated averaging (FedAvg) [16] algorithm was first proposed as a heuristic to improve communication efficiency and data privacy, but later theoretically confirmed to achieve a highly desirable $\mathcal{O}(1/\sqrt{mKT})$ convergence rate in FL (implying linear convergence speedup as the number of clients $m$ increases). Since then, many follow-up works have been proposed to achieve the $\mathcal{O}(1/\sqrt{mKT})$ convergence rate for i.i.d. datasets [17–23] and non-i.i.d. datasets [24–33]. For a comprehensive survey on FL convergence rate order, we refer readers to Section 3 in [34].

**2) Min-max Optimization:** Min-max optimization has a long history dating back to at least [35, 36]. For non-convex-strongly-concave min-max problems, a simple approach is the stochastic gradient descent ascent (SGDA), which performs stochastic gradient descent on primal variables and stochastic gradient ascent on dual variables, respectively. It is well-known that SGDA achieves an $\mathcal{O}(1/\sqrt{T})$ convergence rate [37, 38] for non-convex-strongly-concave min-max problems, matching that of SGD in non-convex optimization. However, in the federated non-convex-strongly-concave setting, studies in [13] and [14] only proved $\mathcal{O}(1/\sqrt{mT})$ and $\mathcal{O}(1/(mKT)^{1/3})$ convergence rates, respectively. So far, it remains unknown whether federated SGDA could achieve the same desirable convergence rate of $\mathcal{O}(1/\sqrt{mKT})$ as FedAvg. In this paper, we show that our SAGDA algorithm and FSGDA (implied by SAGDA) indeed achieve the $\mathcal{O}(1/\sqrt{mKT})$ convergence rate, matching that of FedAvg.

## 3   Problem statement and algorithm design

We consider a general min-max optimization problem in federated learning setting as follows:

$$\min_{\mathbf{x}\in\mathbb{R}^d} \max_{\mathbf{y}\in\mathbb{R}^d} f(\mathbf{x},\mathbf{y}) := \min_{\mathbf{x}\in\mathbb{R}^d} \max_{\mathbf{y}\in\mathbb{R}^d} \frac{1}{M}\sum_{i\in[M]} f_i(\mathbf{x},\mathbf{y}), \tag{1}$$

where $f_i(\mathbf{x},\mathbf{y}) := \mathbb{E}_{\xi_i\sim D_i}[f(\mathbf{x},\mathbf{y},\xi_i)]$ is the local loss function associated with a local data distribution $D_i$ and $M$ is the number of workers. Similar to FL, these exist two main challenges in federated min-max optimization: 1) datasets are generated locally at the clients and generally non-i.i.d., i.e., $D_i \neq D_j$, for $i \neq j$; 2) potentially only a subset of clients may participate in each communication round, leading to partial client participation.

In this paper, we focus on general non-convex-PL min-max problems. Before presenting the algorithms and their convergence analysis, we first state several assumptions.

**Assumption 1.** *(Lipschitz Smooth)* $f_i(\mathbf{x}, \mathbf{y})$ *is* $L_f$-*smooth, i.e., there exists a constant* $L_f > 0$*, so that* $\|\nabla f_i(\mathbf{x}_1, \mathbf{y}_1) - \nabla f_i(\mathbf{x}_2, \mathbf{y}_2)\|^2 \leq L_f^2 \left(\|\mathbf{x}_1 - \mathbf{x}_2\|^2 + \|\mathbf{y}_1 - \mathbf{y}_2\|^2\right), \forall \mathbf{x}_1, \mathbf{x}_2, \mathbf{y}_1, \mathbf{y}_2 \in \mathbb{R}^d, i \in [M].$

**Assumption 2.** *(Polyak-Łojasiewicz (PL) Condition) There exists a constant* $\mu > 0$ *such that* $\forall \mathbf{x}, \mathbf{y}$,

$$\|\nabla_{\mathbf{y}} f(\mathbf{x}, \mathbf{y})\|^2 \geq 2\mu \max_{\mathbf{z}} \left(f(\mathbf{x}, \mathbf{z}) - f(\mathbf{x}, \mathbf{y})\right).$$

Further, we assume the stochastic gradients with respect to $\mathbf{x}$ and $\mathbf{y}$ in each local update step at each client are unbiased and have bounded variances.

**Assumption 3.** *(Unbiased Local Stochastic Gradient) Let* $\xi^i$ *be a random local data sample at client* $i$*. The local stochastic gradients with respect to* $\mathbf{x}$ *and* $\mathbf{y}$ *are unbiased and have bounded variances:*

$$\mathbb{E}[\nabla f_i(\mathbf{x}, \mathbf{y}, \xi_i)] = \nabla f_i(\mathbf{x}, \mathbf{y}), \quad \mathbb{E}\left[\|\nabla_{\mathbf{x}} f_i(\mathbf{x}, \mathbf{y}, \xi_i) - \nabla_{\mathbf{x}} f_i(\mathbf{x}, \mathbf{y})\|^2\right] \leq \sigma_x^2,$$

$$\mathbb{E}\left[\|\nabla_{\mathbf{y}} f_i(\mathbf{x}, \mathbf{y}, \xi_i) - \nabla_{\mathbf{y}} f_i(\mathbf{x}, \mathbf{y})\|^2\right] \leq \sigma_y^2,$$

*where the expectation is taken over local distribution* $D_i$.

To analyze the convergence performance of min-max algorithms, we define a surrogate function $\Phi$ for the global minimization as follows: $\Phi(\mathbf{x}) := \max_{\mathbf{y}} f(\cdot, \mathbf{y})$. We will use $\Phi$ as a metric to measure the performance of an algorithm on min-max problems, and the goal is to find an approximate stationary point of $\Phi$ efficiently. Then, we can conclude from previous works (see Lemma A.5 [39] or Lemma 4.3 [38]) that $\Phi$ is $L$-smooth, where $L := L_f + L_f^2/\mu$.

**Definition 1** (Stationarity). *For a differentiable function* $\Phi$*,* $\mathbf{z}$ *is an* $\epsilon$-*stationary point if* $\|\nabla\Phi(\mathbf{z})\| \leq \epsilon$.

**Definition 2** (Complexity). *The communication and client sample complexity are defined as the total number of rounds and stochastic gradients per client to achieve an* $\epsilon$-*stationary point, respectively.*

### 3.1 The Stochastic Averaging Gradient Descent Ascent (SAGDA) Algorithm

To solve Problem (1), FedAvg could be naturally extended to federated min-max problems by applying SGDA with multiple local update steps in primal and dual variables respectively. However, current results [13–15] show that there exists two limitations: 1) limited data heterogeneity is often assumed, e.g., bounded gradient dissimilarity assumption; 2) communication complexity is unsatisfactory. In this paper, we propose the SAGDA (stochastic sampling averaging gradient descent ascent) algorithm by utilizing the assembly of stochastic gradients from (randomly sampled) clients as control variates to mitigate the effect of data heterogeneity in federated min-max problems. As will be shown later, SAGDA is able to achieve better communication complexity under arbitrary data heterogeneity.

As illustrated in Algorithm 1, SAGDA contains the following two stages:

1. *On the Server Side:* In each communication round, the server initializes the global model $(\mathbf{x}_t, \mathbf{y}_t)$ at $t = 0$ or updates the global model accordingly when $t > 0$ (Line 3). Specifically, for $t > 0$, upon the reception of all returned parameters from round $t - 1$, the server aggregates them using global learning rates $\eta_{x,g}$ and $\eta_{y,g}$ for $\mathbf{x}$ and $\mathbf{y}$, respectively. Then server samples a subset of clients $S_t$ to participate in the training and broadcast the current global model $(\mathbf{x}_t, \mathbf{y}_t)$ to these clients (Line 4). Here, we follow the same common assumption on client participation as in FL: the clients are uniformly sampled without replacement and a fixed-size subset (i.e., $|S_t| = m$) is chosen in each communication round. A key step here is to construct the control variates $(\bar{\mathbf{v}}_x, \bar{\mathbf{v}}_y, \mathbf{v}_{x,i}, \mathbf{v}_{y,i})$ for server and client. Afterwards, the primal and dual variables alongside their control variates are transmitted to each participated client $i \in S_t$ (Line 7).

2. *On the Client Side:* Upon receiving the latest global model $(\mathbf{x}_t, \mathbf{y}_t)$, each client synchronizes its local model (Line 10). Then, each client performs $K$ local updates for $\mathbf{x}$ and $\mathbf{y}$ simultaneously (Line 11). Upon the completion of local computations, the new local model is sent to the server.

We provide two options in SAGDA. First, in each communication round, client and server need to respectively obtain control variates $(\mathbf{v}_{x,i}, \mathbf{v}_{y,i})$ and $(\bar{\mathbf{v}}_x, \bar{\mathbf{v}}_y)$ for "variance reduction" purpose in primal variable $\mathbf{x}$ and dual variable $\mathbf{y}$ (Lines 5 and 6). Option I requires each client to maintain the control variates $(\mathbf{v}_{x,i}, \mathbf{v}_{y,i})$ across rounds locally (Line 12). As a result, $(\bar{\mathbf{v}}_x, \bar{\mathbf{v}}_y)$ are constructed iteratively (Line 5). In Option II, $(\mathbf{v}_{x,i}, \mathbf{v}_{y,i})$ are instantly calcuated by another round of communication, and

**Algorithm 1** The Stochastic Averaging Gradient Descent Ascent (SAGDA) Algorithm.

---
1: **for** $t = 0, \cdots, T - 1$ **do**
2:    **for** Server **do**
3:       Initialize $\mathbf{x}_0, \mathbf{y}_0$ for $t = 0$, or update global model from previous round for $t > 0$:
$$\mathbf{x}_t = \mathbf{x}_{t-1} + \eta_{x,g}\left(\tfrac{1}{m}\sum_{i \in S_{t-1}} \mathbf{x}_{t-1,i}^{K+1} - \mathbf{x}_{t-1}\right),$$
$$\mathbf{y}_t = \mathbf{y}_{t-1} + \eta_{y,g}\left(\tfrac{1}{m}\sum_{i \in S_{t-1}} \mathbf{y}_{t-1,i}^{K+1} - \mathbf{y}_{t-1}\right).$$
4:       Randomly samples a subset $S_t$ of clients with $|S_t| = m$.
5:       **Option I:** Construct sampling averaging $\bar{\mathbf{v}}_x, \bar{\mathbf{v}}_y$ from the return in the previous round:
$$\bar{\mathbf{v}}_x = \bar{\mathbf{v}}_x + \tfrac{1}{M}\sum_{i \in S_{t-1}} \Delta\mathbf{v}_{x,i}, \qquad \bar{\mathbf{v}}_y = \bar{\mathbf{v}}_y + \tfrac{1}{M}\sum_{i \in S_{t-1}} \Delta\mathbf{v}_{y,i}.$$
6:       **Option II:** The server sends current parameters $\mathbf{z}_t := (\mathbf{x}_t, \mathbf{y}_t)$ to clients in $S_t$ and collects stochastic gradients:
$$\mathbf{v}_{x,i} = \nabla_x f_i(\mathbf{z}_t, \xi_{t,i}), \mathbf{v}_{y,i} = \nabla_y f_i(\mathbf{z}_t, \xi_{t,i}),$$
$$\bar{\mathbf{v}}_x = \tfrac{1}{m}\sum_{i \in S_t} \mathbf{v}_{x,i}, \bar{\mathbf{v}}_y = \tfrac{1}{m}\sum_{i \in S_t} \mathbf{v}_{y,i}.$$
7:       Send $(\mathbf{x}_t, \mathbf{y}_t)$ and $(\bar{\mathbf{v}}_x, \bar{\mathbf{v}}_y)$ to each client $i \in S_t$.
8:    **end for**
9:    **for** Each client $i \in S_t$ **do**
10:       Synchronization: $\mathbf{x}_{t,i}^1 = \mathbf{x}_t, \mathbf{y}_{t,i}^1 = \mathbf{y}_t$ and receiving $\bar{\mathbf{v}}_{x,t}, \bar{\mathbf{v}}_{y,t}$.
11:       Local updates ($k \in [K]$):
$$\mathbf{x}_{t,i}^{k+1} = \mathbf{x}_{t,i}^k - \eta_{x,l}\mathbf{v}_{x,i}^k \text{ (cf. Eq. (2) for } \mathbf{v}_{x,i}^k\text{)};$$
$$\mathbf{y}_{t,i}^{k+1} = \mathbf{y}_{t,i}^k + \eta_{y,l}\mathbf{v}_{y,i}^k \text{ (cf. Eq. (3) for } \mathbf{v}_{y,i}^k\text{)};$$
12:       **Option I:**
         Calculate: $\mathbf{v}_{x,i}^{'} = \nabla_x f_i(\mathbf{z}_t, \xi_{t,i}), \mathbf{v}_{y,i}^{'} = \nabla_y f_i(\mathbf{z}_t, \xi_{t,i})$.
         Send $(\mathbf{x}_{t,i}^{K+1}, \mathbf{y}_{t,i}^{K+1})$ and $(\Delta\mathbf{v}_{x,i}, \Delta\mathbf{v}_{y,i}) = \left(\mathbf{v}_{x,i}^{'} - \mathbf{v}_{x,i}, \mathbf{v}_{y,i}^{'} - \mathbf{v}_{y,i}\right)$ to server.
         Assign: $\mathbf{v}_{x,i} = \mathbf{v}_{x,i}^{'}, \mathbf{v}_{y,i} = \mathbf{v}_{y,i}^{'}$.
13:       **Option II:** Send $(\mathbf{x}_{t,i}^{K+1}, \mathbf{y}_{t,i}^{K+1})$ to server.
14:    **end for**
15: **end for**

---

then $(\mathbf{v}_{x,i}, \mathbf{v}_{y,i})$ are constructed accordingly (Line 6). We note that Option I needs client to be *stateful* and thus being more challenging to implement in cross-device FL [34], while Option II may incur extra communication overhead due to the need for one more communication session, although the total communication size remains the same. In the local computation phase, each participated client performs steps (Line 11) based on Eq. (2) and (3), which can be interpreted as "variance reduction." Here, we use $\mathbf{z}_{t,i}^j := (\mathbf{x}_{t,i}^j, \mathbf{y}_{t,i}^j)$ for notational simplicity.

$$\mathbf{v}_{x,i}^k = \nabla_x f_i(\mathbf{z}_{t,i}^k, \xi_{t,i}^k) - \mathbf{v}_{x,i} + \bar{\mathbf{v}}_x, \tag{2}$$

$$\mathbf{v}_{y,i}^k = \nabla_y f_i(\mathbf{z}_{t,i}^k, \xi_{t,i}^k) - \mathbf{v}_{y,i} + \bar{\mathbf{v}}_y. \tag{3}$$

In classic variance reduction methods, the key idea is to utilize a full gradient (or approximation) to reduce the stochastic gradient variance at the expense of high computation complexity compared to SGD. Note that, in federated learning, the gradient dissimilarity (due to data heterogeneity) is a crtical challenge and more problematic than stochastic gradient variance. Therefore, we calculate a 2-tuple $(\bar{\mathbf{v}}_{x,t}, \bar{\mathbf{v}}_{y,t})$ of stochastic gradients from all clients as control variates to mitigate the potential gradient deviation due to data heterogeneity. Note that SAGDA does *not* require a full gradient calculation for each client. With the help from the local steps in (2) and (3), each client no longer generate large deviation in local updates even with *arbitrary* data heterogeneity. The reason is that, for small local learning rates, the local steps in each client could be approximated by

$$\nabla_x f_i(\mathbf{z}_{t,i}^j, \xi_{t,i}^k) \approx \mathbf{v}_{x,i} \implies \mathbf{v}_{x,i}^k \approx \bar{\mathbf{v}}_x,$$

$$\nabla_y f_i(\mathbf{z}_{t,i}^k, \xi_{t,i}^k) \approx \mathbf{v}_{y,i} \implies \mathbf{v}_{y,i}^k \approx \bar{\mathbf{v}}_y.$$

In other words, SAGDA mimics mini-batch SGDA in the centralized learning by using an approximation of mini-batch stochastic gradient for the updates. As a result, SAGDA is able to provide a

desirable convergence rate, while allowing arbitrary data heterogeneity. We state the convergence rate result of SAGDA as follows:

**Theorem 1** (Convergence Rate of SAGDA). *Under Assumptions 1- 3, define $\mathcal{L}_t = \Phi(\mathbf{x}_t) - \frac{1}{10}f(\mathbf{x}_t, \mathbf{y}_t)$, the output sequence $\{\mathbf{x}_t\}$ generated by* SAGDA *satisfies:*

• *For Option I with learning rates $\eta_{x,g}$, $\eta_{x,l}$, $\eta_{y,g}$, and $\eta_{y,l}$ satisfying*

$$8K(K-1)(2K-1)L_f^2 \max\{\eta_{x,l}^2, \eta_{y,l}^2\} \leq 1,$$

$$\frac{1}{2} - 4a_2 L_f^2 K^2 \left(\eta_x^2 + \eta_y^2\right) - \left(a_1 + a_2 4 L_f^2 K^2 \left(\eta_x^2 + \eta_y^2\right)\right) 160K^2 \left(\eta_{x,l}^2 + \eta_{y,l}^2\right) L_f^2 \geq 0,$$

$$\left[\frac{1}{10}\eta_x K - 4a_2 K^2 \eta_x^2\right] - \left[a_1 + a_2 4 L_f^2 K^2 \left(\eta_x^2 + \eta_y^2\right)\right] 40K^2 \eta_{x,l}^2 \geq 0,$$

$$\left[\eta_y K \left(\frac{1}{20} - \frac{\eta_x}{\eta_y}\frac{L_f^2}{\mu^2}\right) - 4a_2 K^2 \eta_y^2\right] - \left[a_1 + a_2 4 L_f^2 K^2 \left(\eta_x^2 + \eta_y^2\right)\right] 40K^2 \eta_{y,l}^2 \geq 0,$$

*where $a_1 = K L_f^2 \left(\frac{31}{20}\eta_x + \frac{1}{20}\eta_y\right)$ and $a_2 = \frac{1}{2}\left(L + \frac{L_f}{10}\right) + 1 + \frac{M^2}{m^2} - \frac{M}{m}$, it holds that*

$$\frac{1}{T}\sum_{t=0}^{T-1}\mathbb{E}\|\nabla\Phi(\mathbf{x}_t)\|^2 \leq \underbrace{\frac{2(\mathcal{L}_0 - \mathcal{L}_*)}{\eta_x K T}}_{\text{optimization error}} + \underbrace{\left[\left(L + \frac{L_f}{10}\right) + 4\right]\frac{9}{m\eta_x}\left(\eta_x^2\sigma_x^2 + \eta_y^2\sigma_y^2\right)}_{\text{statistical error}} + \underbrace{\psi_1}_{\text{local update error}}$$

*where $\psi_1$ is defined as follows:*

$$\psi_1 = \left[L_f^2\left(\frac{31}{20} + \frac{1}{20}\frac{\eta_y}{\eta_x}\right) + \left[\frac{1}{2}\left(L + \frac{L_f}{10}\right) + 2\right]4L_f^2 K\left(\eta_x + \frac{\eta_y^2}{\eta_x}\right)\right]\left[20K^2\left(\eta_{x,l}^2\sigma_x^2 + \eta_{y,l}^2\sigma_y^2\right)\right].$$

• *For Option II with learning rates $\eta_{x,g}$, $\eta_{x,l}$, $\eta_{y,g}$, and $\eta_{y,l}$ satisfying*

$$8K(K-1)(2K-1)L_f^2 \max\{\eta_{x,l}^2, \eta_{y,l}^2\} \leq 1,$$

$$\frac{1}{10}\eta_x K - \left(2\left(L + \frac{L_f}{10}\right)\eta_x^2 K^2 + 40K^2\eta_{x,l}^2 b_1\right) \geq 0,$$

$$\eta_y K\left(\frac{1}{20} - \frac{\eta_x}{\eta_y}\frac{L_f^2}{\mu^2}\right) - \left(\frac{1}{5}L_f\eta_y^2 K^2 + 40K^2\eta_{y,l}^2 b_1\right) \geq 0,$$

*where $b_1 = L_f^2\left[\frac{31}{20}\eta_x K + \frac{1}{20}\eta_y K + 2\left(L + \frac{L_f}{10}\right)\eta_x^2 K^2 + \frac{1}{5}L_f\eta_y^2 K^2\right]$, it holds that*

$$\frac{1}{T}\sum_{t=0}^{T-1}\mathbb{E}\|\nabla\Phi(\mathbf{x}_t)\|^2 \leq \frac{2(\mathcal{L}_0 - \mathcal{L}_*)}{\eta_x K T} + \left[\left(L + \frac{L_f}{10}\right)\frac{9\eta_x}{m}\sigma_x^2 + + \frac{9}{10}L_f\frac{\eta_y^2}{m\eta_x}\sigma_y^2\right] + \psi_2.$$

*where $\psi_2$ is defined as follows:*

$$\psi_2 = L_f^2\left[\frac{31}{20}K + \frac{1}{20}\frac{\eta_y}{\eta_x}K + 2\left(L + \frac{L_f}{10}\right)\eta_x K^2 + \frac{1}{5}L_f\frac{\eta_y^2}{\eta_x}K^2\right]\left[10(16K + 1)\right]\left(\eta_{x,l}^2\sigma_x^2 + \eta_{y,l}^2\sigma_y^2\right).$$

Here $\eta_x = \eta_{x,l}\eta_{x,g}$ and $\eta_y = \eta_{y,l}\eta_{y,g}$. The convergence rate results in Theorem 1 contain three terms: optimization error, statistical error and local update error. The first two errors are similar to those in first-order stochastic methods, which are optimization errors due to initial point and statistical error originated from stochastic gradient variance. The local updates without synchronization among clients result in deviations that contribute to the third error. For the learning rates, if we use a sufficiently small local learning rates $\eta_{x,l}$ and $\eta_{y,l}$, it requires that $\eta_x K = \mathcal{O}(1)$ and $\eta_y K = \mathcal{O}(1)$.

Based on Theorem 1, we immediately have the following result:

---

**Algorithm 2** Federated Stochastic Gradient Descent Ascent (FSGDA) Algorithm.

---
1: **for** $t = 0, \cdots, T-1$ **do**
2:    **for** Server **do**
3:       Initialize $(\mathbf{x}_0, \mathbf{y}_0)$ for $t = 0$, or update global model from previous round for $t > 0$:

$$\mathbf{x}_t = \mathbf{x}_{t-1} + \eta_{x,g} \left( \frac{1}{m} \sum_{i \in S_{t-1}} \mathbf{x}_{t-1,i}^{K+1} - \mathbf{x}_{t-1} \right), \mathbf{y}_t = \mathbf{y}_{t-1} + \eta_{y,g} \left( \frac{1}{m} \sum_{i \in S_{t-1}} \mathbf{y}_{t-1,i}^{K+1} - \mathbf{y}_{t-1} \right).$$

4:       The server randomly samples a subset $S_t$ of clients with $|S_t| = m$ and sends current parameters $(\mathbf{x}_t, \mathbf{y}_t)$.
5:    **end for**
6:    **for** Each client $i \in S_t$ **do**
7:       Synchronization: $\mathbf{x}_{t,i}^1 = \mathbf{x}_t, \mathbf{y}_{t,i}^1 = \mathbf{y}_t$.
8:       Local updates ($k \in [K]$):

$$\mathbf{x}_{t,i}^{k+1} = \mathbf{x}_{t,i}^k - \eta_{x,l} \nabla_{\mathbf{x}} f_i(\mathbf{x}_{t,i}^k, \mathbf{y}_{t,i}^k, \xi_{t,i}^k), \quad \mathbf{y}_{t,i}^{k+1} = \mathbf{y}_{t,i}^k + \eta_{y,l} \nabla_{\mathbf{y}} f_i(\mathbf{x}_{t,i}^k, \mathbf{y}_{t,i}^k, \xi_{t,i}^k).$$

9:       Send $\left(\mathbf{x}_{t,i}^{K+1}, \mathbf{y}_{t,i}^{K+1}\right)$ to server.
10:   **end for**
11: **end for**

---

**Corollary 1** (Convergence Rate of SAGDA ). *Let* $\eta_x = \Theta(\frac{\sqrt{m}}{\sqrt{KT}}), \eta_x = \Theta(\frac{\sqrt{m}}{\sqrt{KT}})$, $\eta_{x,l} \leq \min\{\frac{1}{m^{1/2}K^{3/2}}, \frac{K^{3/4}}{m^{1/4}T^{1/4}}\}, \eta_{y,l} \leq \min\{\frac{1}{m^{1/2}K^{3/2}}, \frac{K^{3/4}}{m^{1/4}T^{1/4}}\}$, *and* $T = \Omega(mK)$, *the convergence rate of* SAGDA *is* $\mathcal{O}(\frac{1}{\sqrt{mKT}})$.

Corollary 1 says that, for sufficiently many communication rounds ($T = \Omega(mK)$), SAGDA achieves the linear speedup in both $m$ and $K$. In other words, the per-client sample complexity and communication complexity are $\mathcal{O}((1/m)\epsilon^{-4})$ and $\mathcal{O}(\epsilon^{-2})$, respectively. The per-client sample complexity indicates the benefits of parallelism as the number of clients $m$ increases. The communication complexity significantly improves those in existing works by at least a $(1/\epsilon)$-factor (cf. Table 1).

### 3.2 Special case of SAGDA: Federated stochastic gradient descent ascent (FSGDA)

We note that, if we set all the control variates to zero, SAGDA reduces to the federated stochastic gradient descent ascent (FSGDA) method, which is a natural extension of FedAvg and SGDA to federated min-max learning[1]. We show that much improved convergence rate results of FSGDA can be directly implied by SAGDA.

For a fair comparison with existing works, we also adopt the same bounded gradient dissimilarity assumption as in [14, 15] to bound the second moment between gradients of local and global loss functions (i.e., quantifying data heterogeneity).

**Assumption 4.** *(Bounded Gradient Dissimilarity) There exist two constants* $\sigma_{x,G} \geq 0$ *and* $\sigma_{y,G} \geq 0$ *such that* $\mathbb{E}\left[\|\nabla_x f_i(\mathbf{x}, \mathbf{y}) - \nabla_x f(\mathbf{x}, \mathbf{y})\|^2\right] \leq \sigma_{x,G}^2$ *and* $\mathbb{E}\left[\|\nabla_y f_i(\mathbf{x}, \mathbf{y}) - \nabla_y f(\mathbf{x}, \mathbf{y})\|^2\right] \leq \sigma_{y,G}^2$.

Assumption 4 is a commonly-used assumption to quantify the data heterogeneity [14, 15]. Based on the results and analysis of SAGDA , we can show the following convergence results for FSGDA:

**Theorem 2** (Convergence Rate for FSGDA). *Under Assumptions 1- 4, define* $\mathcal{L}_t = \Phi(\mathbf{x}_t) - \frac{1}{10} f(\mathbf{x}_t, \mathbf{y}_t)$, *if the learning rates* $\eta_{x,g}$, $\eta_{x,l}$, $\eta_{y,g}$, *and* $\eta_{y,l}$ *satisfy:*

$$8K(K-1)(2K-1)L_f^2 \max\{\eta_{x,l}^2, \eta_{y,l}^2\} \leq 1,$$

$$a_1 - a_3 40 L_f^2 K^2 \eta_{x,l}^2 - \frac{\eta_y}{\eta_x} a_4 40 L_f^2 K^2 \eta_{x,l}^2 \geq 0,$$

$$a_2 - a_3 \frac{\eta_x}{\eta_y} 40 L_f^2 K^2 \eta_{y,l}^2 - a_4 40 L_f^2 K^2 \eta_{y,l}^2 \geq 0,$$

---
[1]Our FSGDA is in fact a generalized version of local FSGDA [14, 15] as our FSGDA has two-sided learning rates and client sampling. If $\eta_{x,g} = \eta_{y,g} = 1$, our FSGDA is exactly the same as the standard FSGDA.

where $a_1 = \left(\frac{1}{10} - 2(2L + \frac{1}{5}L_f)\eta_x K\right), a_2 = \left(\frac{1}{20} - \frac{2}{5}L_f\eta_y K - \frac{\eta_x}{\eta_y}\frac{L_f^2}{\mu^2}\right), a_3 = \left(\frac{31}{20} + (2L + \frac{1}{5}L_f)\eta_x K\right)$ and $a_4 = \left(\frac{1}{20} + \frac{1}{5}L_f\eta_y K\right)$, then the output sequence $\{\mathbf{x}_t\}$ generated by FSGDA satisfies:

$$\frac{1}{T}\sum_{t=0}^{T-1}\mathbb{E}\|\nabla\Phi(\mathbf{x}_t)\|^2 \leq \underbrace{\frac{2\left(\mathcal{L}_0 - \mathcal{L}_T\right)}{\eta_x KT}}_{\text{optimization error}} + \underbrace{\frac{2\eta_x}{m}\left(L + \frac{L_f}{100}\right)\sigma_x^2 + \frac{L_f\eta_y^2}{5m\eta_x}\sigma_y^2}_{\text{statistical error}} + \underbrace{\psi_3}_{\substack{\text{local} \\ \text{update error}}} + \underbrace{\psi_4}_{\substack{\text{sampling} \\ \text{variance}}}.$$

Here, $\psi_3$ and $\psi_4$ are defined as follows:

$$\psi_3 = 2\left(a_3 L_f^2 + a_4\frac{\eta_y}{\eta_x}L_f^2\right)\left[40K^2\eta_{x,l}^2\sigma_{x,G}^2 + 40K^2\eta_{y,l}^2\sigma_{y,G}^2 + 5K\eta_{x,l}^2\sigma_x^2 + 5K\eta_{y,l}^2\sigma_y^2\right],$$

$$\psi_4 = \left((2L + \frac{1}{5}L_f)\eta_x K\right)\left(1 - \frac{m}{M}\right)\frac{2}{m}\sigma_{x,G}^2 + \frac{2}{5m}L_f\eta_y K\frac{\eta_y}{\eta_x}\left(1 - \frac{m}{M}\right)\sigma_{y,G}^2,$$

The convergence rate result in Theorem 2 contains four parts. The first three terms are similar to the errors in SAGDA analysis. However, one difference is that the local update error heavily depends on the data heterogeneity parameter $\sigma_{x,G}$ and $\sigma_{y,G}$. Specifically, the error grows at least linearly with respect to the local step in terms stochastic variance $\sigma_x^2$ and $\sigma_y^2$ and quadratically in the gradient dissimilarity $\sigma_{x,G}^2$ and $\sigma_{y,G}^2$. This indicates that data heterogeneity is more problematic than stochastic gradient variance, yielding a larger error in local updates and thus necessitating smaller local steps. Fortunately, this error is associated with the square of local learning rates $\eta_{x,l}^2$ and $\eta_{y,l}^2$. So, with sufficiently small local learning rates, $\psi_1$ can be easily reduced. In other words, under bounded gradient dissimilarity (i.e., data heterogeneity), small local learning rates render controllable local update error among clients.

Partial client participation by random sampling without replacement is an unbiased estimation of the global loss function and has a bounded variance, contributing to the third term $\psi_4$. Will full clients participation ($m = M$), this error term can be reduced to zero through our analysis.

Theorem 2 implies a *new* result for FSGDA: if we use sufficiently small local learning rates under full client participation, FSGDA achieves a similar convergence rate to those of SGD and SGDA:

**Corollary 2** (Convergence Rate of FSGDA under Full Client Participation). *Considering full client participation ($m = M$), let $\eta_x = \Theta(\frac{\sqrt{m}}{\sqrt{KT}}), \eta_x = \Theta(\frac{\sqrt{m}}{\sqrt{KT}})$, $\eta_{x,l} \leq \frac{1}{(mT)^{1/4}K^{5/4}}, \eta_{y,l} \leq \frac{1}{(mT)^{1/4}K^{5/4}}$, and $T = \Omega(mK)$, the convergence rate of FSGDA algorithm is: $\mathcal{O}(\frac{1}{\sqrt{MKT}})$.*

This convergence rate indicates the linear speedup effect in terms of both $M$ and $K$. However, we note that this is subject to learning rates constraints, which does not allow arbitrarily many local steps. Specifically, the number of local step in FSGDA is on the order of $K = \mathcal{O}(T/M)$. Hence, FSGDA achieves per-client sample complexity $\mathcal{O}(\frac{1}{m\epsilon^4})$ and communication complexity $\mathcal{O}(\frac{1}{\epsilon^2})$. We improve the state-of-the-art communication comoplexity from $\mathcal{O}(\frac{1}{\epsilon^3})$ in [15] to $\mathcal{O}(\frac{1}{\epsilon^2})$ in our paper.

For partial client participation $m < M$, however, FSGDA can only have the following convergence rate under appropriate learning rates.

**Corollary 3** (Convergence Rate of FSGDA under Partial Client Participation). *Let $\eta_x = \Theta(\frac{\sqrt{m}}{\sqrt{T}K}), \eta_y = \Theta(\frac{\sqrt{m}}{\sqrt{T}K})$, $\eta_{x,l} \leq \frac{1}{(mT)^{1/4}K}$, and $\eta_{y,l} \leq \frac{1}{(mT)^{1/4}K}$, the convergence rate of FSGDA algorithm is $\mathcal{O}(1/\sqrt{mT})$.*

For partial client participation, we note that only the linear speedup in $m$ is achievable and the linear speedup in $K$ is not achievable due to the impact of sampling variance. According to previous works in federated minimization problem, the convergence bounds also have this observation [28, 31]. To our knowledge, we are not aware of any existing results on linear speedup in $K$ with partial client participation. We will leave this as an open problem in our future studies.

## 4 Numerical Experiments

In this section, we conduct numerical experiments using two machine learning problems (Logistic Regression and AUC Maximization) to verify our theoretical results for SAGDA as well as FSGDA.

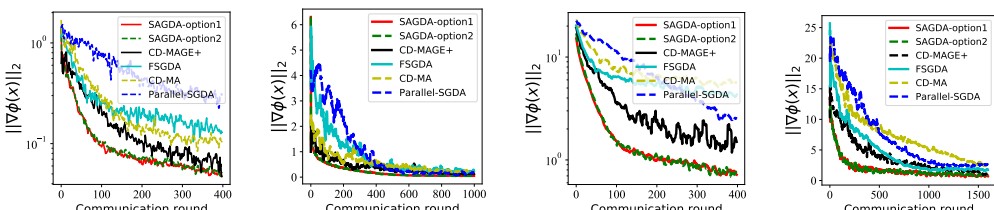

(a) Logistic regression under "a9a" dataset.  (b) Logistic regression under "MNIST" dataset.  (c) AUC maximization under "a9a" dataset.  (d) AUC maximization under "MNIST" dataset.

Figure 1: Comparisons of federated min-max learning algorithms in terms of communication rounds.

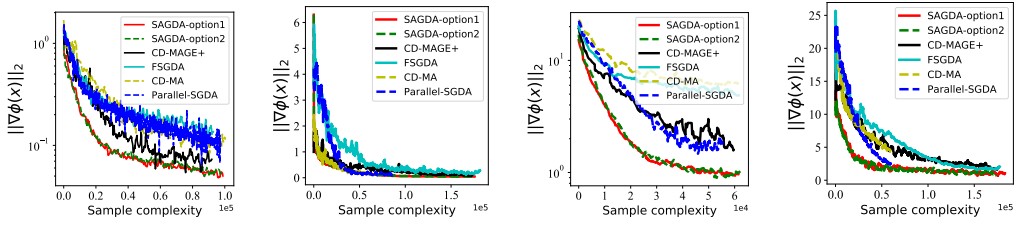

(a) Logistic regression under "a9a" dataset.  (b) Logistic regression under "MNIST" dataset.  (c) AUC maximization under "a9a" dataset.  (d) AUC maximization under "MNIST" dataset.

Figure 2: Comparisons of federated min-max learning algorithms in terms of sample complexity.

Due to space limitation, detailed discriptions for machine learning models and additional experiments for parameter tuning are relegated to our supplementary material.

We compare our algorithms using Parallel-SGDA [14, 15, 38], CD-MA [13], and CD-MAGE+ [13] as baselines in our experiments. We note that the CD-MAGE+ method is the state-of the-art federated minimax algorithm.

- *Parallel-SGDA [14, 15, 38]:* Parallel-SGDA is the parallel version of the stochastic gradient descent ascent(SGDA) algorithm. Each agent $i$ updates its local parameters as: $\mathbf{x}_{t+1,i} = \mathbf{x}_{t,i} - \eta_{x,l}\nabla_x f_i(\mathbf{x}_{t,i}, \mathbf{y}_{t,i}, \xi_{t,i})$ and $\mathbf{y}_{t+1,i} = \mathbf{y}_{t,i} - \eta_{y,l}\nabla_y f_i(\mathbf{x}_{t,i}, \mathbf{y}_{t,i}, \xi_{t,i})$.
- *CD-MA [13]:* Each agent $i$ updates its local parameters with mini-batch estimators. The server computes $\mathbf{x}_{t+1} = \mathbf{x}_t + \frac{1}{m}\sum_{i \in S_t} \mathbf{x}_{t,i}^{K+1}$, $\mathbf{y}_{t+1} = \mathbf{y}_t + \frac{1}{m}\sum_{i \in S_t} \mathbf{y}_{t,i}^{K+1}$, where $S_t$ is the subsets of clients.
- *CD-MAGE+ [13]:* Each agent $i$ updates its local parameters with a recursive momentum-based estimator. The server does the same procedure as in CD-MA.

**1) Datasets:** We test the convergence performance of our algorithms using the "a9a" dataset [40] and "MNIST" [41] from LIBSVM repository. The "a9a'' readily contains two categories for classification. To generate data with two categories for "MNIST", we split it into two classes by treating the number "1" class as the positive class and the remaining as the negative class. We randomly selected 5000 data points from the positive class and 5000 data points from the negative class in the data repository. To generate heterogeneous data, the training data is first sorted according to the original class label and then equally partitioned into 100 workers so that all data points on one client are from the same class.

**2) Parameter Settings:** We initialize all algorithms at the same point, generated randomly from the random number generator in Python. The learning rates are chosen as $\eta_{x,l} = \eta_{y,l} = 10^{-2}, \eta_{x,g} = \eta_{y,g} = 2$, local updates $K = 10$. We have $m = 100$ clients and each client has $n = 100$ samples.

**3) Performance Comparisons:** As shown in Fig. 1, we conduct experiments by using distributionally robust optimization with non-convex regularized logistic loss and by AUC maximization on both "a9a" and "MNIST" datasets. We compare the convergence results in terms of the number of communication rounds and sample complexity. For better visualization, the results are smoothed by averaging the values over a window of size five. It can be seen from Fig. 2 that SAGDA converges

faster than the baseline algorithms (CD-MA, CD-MAGE+, Parallel-SGDA) in terms of the total number of communication rounds. We can also observe that SAGDA have a lower sample complexity than all the other algorithms. As mentioned in Section. 3, the local learning rates are necessarily small since they are used to control the local update errors. Thanks to the relatively large step-size $\eta_{x,g}, \eta_{y,g}$ in SAGDA, the actual learning rate $\eta_x = \eta_{x,l} \times \eta_{x,g}$ and $\eta_y = \eta_{y,l} \times \eta_{y,g}$ help us achieve better convergence performance in each communication round. Our experimental results thus verify our theoretical analysis that SAGDA is able to achieve both low sample and communication complexities.

## 5 Conclusion

In this paper, we considered federated min-max learning with the goal of achieving low communication complexity. We proposed a new algorithmic framework called SAGDA, which i) assembles stochastic gradient estimators from randomly sampled clients as control variates and ii) leverages two learning rates on both server and client sides. We showed that SAGDA achieves a linear speedup in terms of both the number of clients and local update steps, which yields an $\mathcal{O}(\epsilon^{-2})$ communication complexity that is orders of magnitude lower than the state of the art. Also, by noting that the standard federated stochastic gradient descent ascent (FSGDA) is in fact a special case of SAGDA, we also obtained an $\mathcal{O}(\epsilon^{-2})$ communication complexity result for FSGDA. Extensive numerical experiments corroborated the effectiveness and efficiency of our algorithms.

## Acknowledgements

This work has been supported in part by NSF grants CAREER CNS-2110259, CNS-2112471, ECCS-2140277, and CCF-2110252.

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
