# A    Further Experiments and Additional Results

In the following, we provide the detailed machine learning models for our experiments:

**1) Logistic Regression Model:** We use the following min-max regression problem with datasets $\xi_i := \{(\mathbf{a}_{ij}, b_{ij})\}_{j=1}^n$, where $\mathbf{a}_{ij} \in \mathbb{R}^d$ is the feature of the $j$-th sample of worker $i$ and $b_{ij} \in \{1, -1\}$ is the associated label:

$$\min_{\mathbf{x} \in \mathbb{R}^d} \max_{\mathbf{y} \in \mathbb{R}^n} \frac{1}{m} \sum_{i \in M} f_i(\mathbf{x}, \mathbf{y}),$$

where $f_i(\mathbf{x}, \mathbf{y})$ is defined as:

$$f_i(\mathbf{x}, \mathbf{y}) \triangleq \frac{1}{n} \sum_{j=1}^n [y_j l_j(\mathbf{x}) - V(\mathbf{y}) + g(\mathbf{x})], \tag{4}$$

where the loss function $l_i(\mathbf{x}) \triangleq \log\left(1 + \exp\left(-b_{ij}\mathbf{a}_{ij}^\top\mathbf{x}\right)\right), g(\mathbf{x}) \triangleq \lambda_2 \sum_{k=1}^d \frac{\alpha x_k^2}{1+\alpha x_k^2}$, and $V(\mathbf{y}) = \frac{1}{2}\lambda_1\|n\mathbf{y} - \mathbf{1}\|_2^2$. We choose constants $\lambda_1 = 1/n^2$, $\lambda_2 = 10^{-3}$ and $\alpha = 10$.

**2) AUC Maximization:** We use a dataset $\{\mathbf{a}_{ij}, b_{ij}\}_{j=1}^n$, where $\mathbf{a}_{ij} \in \mathbb{R}^d$ is the feature of the $j$-th sample of worker $i$, $\mathbf{w}_i$ denotes a feature vector and $b_{ij} \in \{-1, +1\}$ denotes the corresponding label. For a scoring function $h_\mathbf{x}$ of a classification model parameterized by $\mathbf{x} \in \mathbb{R}^d$, the AUC maximization problem is defined as:

$$\max_\mathbf{x} \frac{1}{m^+ m^-} \sum_{b_{ij}=+1, b_{ik}=-1} \mathbb{I}_{\{h_\mathbf{x}(\mathbf{a}_{ij}) \geq h_\mathbf{x}(\mathbf{a}_{ik})\}}, \tag{5}$$

where $m^+$ denotes the number of positive samples, $m^-$ denotes the number of negative samples, and $\mathbb{I}_{\{\cdot\}}$ represents the indicator function. The above optimization problem can be reformulated as the following min-max optimization problem [1, 2]:

$$\min_{(\mathbf{x},c_1,c_2)\in\mathbb{R}^{d+2}} \max_{\lambda\in\mathbb{R}} f(\mathbf{x}, c_1, c_2, \lambda)$$

$$:= \frac{1}{mn} \sum_{i\in M} \sum_{j=1}^n \{(1-\tau)(h_\mathbf{x}(\mathbf{a}_{ij})-c_1)^2\mathbb{I}_{\{b_{ij}=1\}} - \tau(1-\tau)\lambda^2$$

$$+ \tau(h_\mathbf{x}(\mathbf{a}_{ij})-c_2)^2\mathbb{I}_{\{b_{ij}=-1\}} + 2(1+\lambda)\tau h_\mathbf{x}(\mathbf{a}_{ij})\mathbb{I}_{\{b_{ij}=-1\}}$$

$$- 2(1+\lambda)(1-\tau)h_\mathbf{x}(\mathbf{a}_{ij})\mathbb{I}_{\{b_{ij}=1\}}, \}, \tag{6}$$

where $\tau := m^+ / (m^+ + m^-)$ is the fraction of positive data. Note that $f(\mathbf{x}, c_1, c_2, \cdot)$ is strongly concave for any $(\mathbf{x}, c_1, c_2) \in \mathbb{R}^{d+2}$.

**3) Generator Adverserial Networks(GANs):** Although our paper is focused on general non-convex-PL min-max problems, we believe that our paper will benefit from comparing further experimental results on the convergence performance of nonconvex-nonconcave problems (e.g., GANs), since the non-convex-PL problem is a special case for nonconvex-nonconcave min-max problems.

In our experiment, generator network is parameterized by $\mathbf{x}$ as $G_\mathbf{X}$ and the discriminator network parameterized by $\mathbf{y}$ as $D_\mathbf{y}$. We adopt the following loss function:

$$f_i(\mathbf{x}, \mathbf{y}) = \mathbb{E}_{\mathbf{a}_i\sim\mathcal{P}_{true}}[\log D_\mathbf{y}(\mathbf{a}_i)] + \mathbb{E}_{\mathbf{z}\sim\mathcal{P}_\mathbf{z}}[\log(1 - D_\mathbf{y}(G_\mathbf{x}(\mathbf{z})))]$$

where $\mathbf{a}_i$ is the data point on client $i$ and $\mathcal{P}_{true}$ is the distribution of the true samples. $z$ denotes the input noise vector and $\mathcal{P}_z$ is the prior distribution of the noise vector for generating samples. We have tested the convergence performance of our algorithms using the MNIST dataset. We chose the learning rates as $\eta_{x,l} = \eta_{y,l} = 10^{-2}, \eta_{x,g} = \eta_{y,g} = 2$, local updates

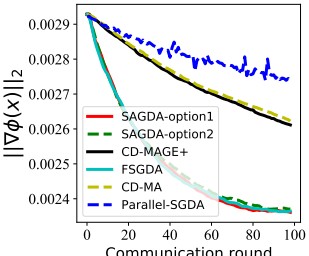

Figure 3:    GANs under "MNIST" dataset.

$K = 10$. We have $m = 100$ clients and each client has $n = 100$ samples. Again, from Fig. 3, we can observe that both our proposed algorithms FSGDA and FSGDA have better convergence performance compared with the baselines.

**Impact of the Local Steps:** In this section, we run additional experiments to investigate the impact of the local steps $K$ on the training performance. We run FSGDA and SAGDA over the hetergenous "a9a" [40] dataset with the regression model mentioned in Section 4. We fix the local step-size at 0.01, worker number at 100, and choose the number of local update rounds $K$ from the discrete set $\{2, 10, 20\}$. In terms of communication round, the gradient norm $\|\nabla\phi(\mathbf{x})\|^2$ decreases as $K$ increases. This is due to the fact that the algorithm needs more communication round while $K$ is small, which matches our Corollary 2 and Corollary 3.

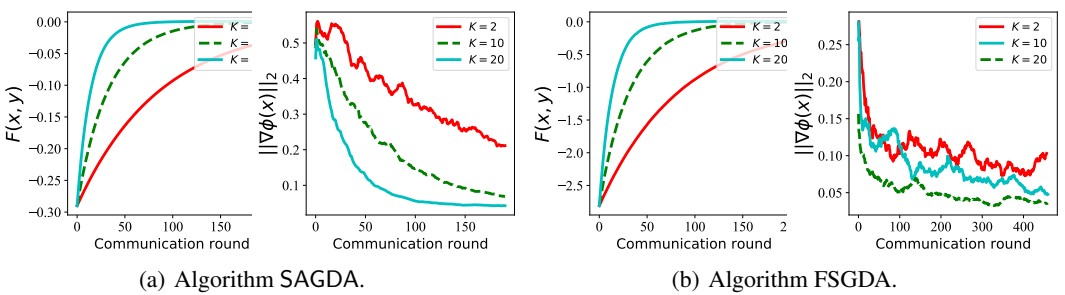

(a) Algorithm SAGDA.         (b) Algorithm FSGDA.

Figure 4: Algorithm performance under different local $K$ steps.

**Impact of the Local Step-size:** In this experiment, we choose the value of the local step-sizes from the discrete set $\{0.0001, 0.001, 0.01\}$ and fix worker number at 100, local update rounds at 10. As shown in Fig. 5(a) and Fig.6(a) , larger local step-sizes lead to faster convergence rates.

**Impact of the Global Step-size:** we choose the global step-sizes value from the discrete set $\{2, 5, 10\}$ and fix worker number at 100, local update rounds at 10. As shown in Fig. 5(b) and 6(b) and, larger global step-sizes lead to faster convergence rates.

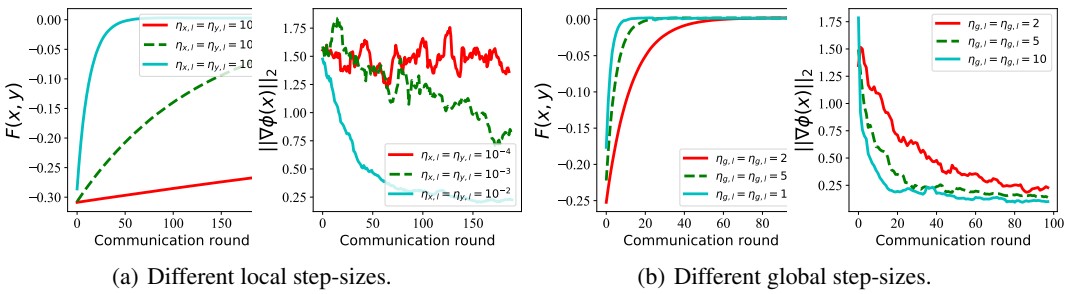

(a) Different local step-sizes.       (b) Different global step-sizes.

Figure 5: The FSGDA algorithm under different step-sizes.

# B Proof

## B.1 Proof for FSGDA

For notational simplicity and clarity, we have the following definitions.

$$\Phi(\mathbf{x}) = \max_{\mathbf{y}\in\mathbb{R}^d} f(\mathbf{x}, \mathbf{y});$$

$$\mathbf{z}_t = (\mathbf{x}_t, \mathbf{y}_t);$$

$$\eta_x = \eta_{x,g}\eta_{x,l}, \eta_y = \eta_{y,g}\eta_{y,l};$$

$$\mathbf{u}_{x,t} = \frac{1}{m}\sum_{i\in S_t}\nabla_x f_i(\mathbf{z}_t), \mathbf{u}_{y,t} = \frac{1}{m}\sum_{i\in S_t}\nabla_y f_i(\mathbf{z}_t).$$

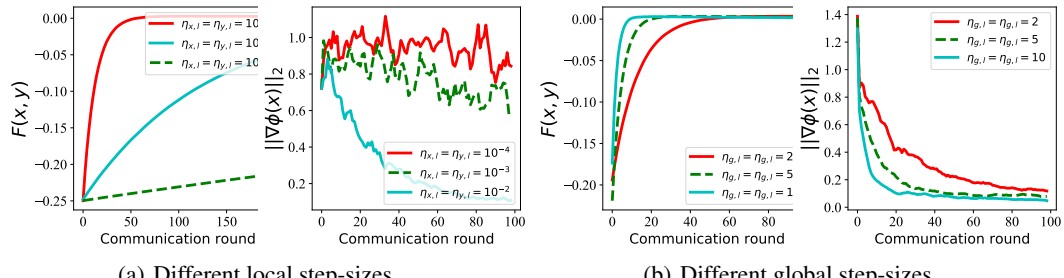

(a) Different local step-sizes.  (b) Different global step-sizes.

Figure 6: The SAGDA algorithm under different step-sizes.

For simplicity, we write the update step uniformly:

$$\mathbf{x}_{t+1} = \mathbf{x}_t - \eta_x K(\mathbf{u}_{x,t} - \mathbf{e}_{x,t}),$$
$$\mathbf{y}_{t+1} = \mathbf{y}_t + \eta_y K(\mathbf{u}_{y,t} - \mathbf{e}_{y,t}).$$

For FSGDA , the update rule is:

$$\mathbf{x}_{t+1} = \mathbf{x}_t - \eta_{x,g}\eta_{x,l}\left(\frac{1}{m}\sum_{i\in S_t}\sum_{j\in[K]}\nabla_x f_i(\mathbf{z}_{t,i}^j, \xi_{t,i}^j)\right),$$

$$\mathbf{y}_{t+1} = \mathbf{y}_t + \eta_{y,g}\eta_{y,l}\left(\frac{1}{m}\sum_{i\in S_t}\sum_{j\in[K]}\nabla_y f_i(\mathbf{z}_{t,i}^j, \xi_{t,i}^j)\right),$$

$$\mathbf{e}_{x,t} = \frac{1}{mK}\sum_{i\in S_t}\sum_{j\in[K]}\left(\nabla_x f_i(\mathbf{z}_t) - \nabla_x f_i(\mathbf{z}_{t,i}^j, \xi_{t,i}^j)\right),$$

$$\bar{\mathbf{e}}_{x,t} = \mathbb{E}[\mathbf{e}_{x,t}] = \frac{1}{mK}\sum_{i\in S_t}\sum_{j\in[K]}\left(\nabla_x f_i(\mathbf{z}_t) - \nabla_x f_i(\mathbf{z}_{t,i}^j)\right),$$

$$\mathbf{e}_{y,t} = \frac{1}{mK}\sum_{i\in S_t}\sum_{j\in[K]}\left(\nabla_y f_i(\mathbf{z}_t) - \nabla_y f_i(\mathbf{z}_{t,i}^j, \xi_{t,i}^j)\right),$$

$$\bar{\mathbf{e}}_{y,t} = \mathbb{E}[\mathbf{e}_{y,t}] = \frac{1}{mK}\sum_{i\in S_t}\sum_{j\in[K]}\left(\nabla_y f_i(\mathbf{z}_t) - \nabla_y f_i(\mathbf{z}_{t,i}^j)\right).$$

Note the above expectation is only on the stochastic noise.

**Lemma 1.**

$$\mathbb{E}\left\|\Delta\mathbf{x}_t\right\|^2 = \mathbb{E}\|\left(\mathbf{u}_{x,t} - \mathbf{e}_{x,t}\right)\|^2 \leq 4\mathbb{E}\left\|\bar{\mathbf{e}}_{x,t}\right\|^2 + 4\mathbb{E}\left\|\mathbf{u}_{x,t}\right\|^2 + \frac{2}{mK}\sigma_x^2,$$

$$\mathbb{E}\left\|\Delta\mathbf{x}_t\right\|^2 = \mathbb{E}\|\left(\mathbf{u}_{y,t} - \mathbf{e}_{y,t}\right)\|^2 \leq 4\mathbb{E}\left\|\bar{\mathbf{e}}_{y,t}\right\|^2 + 4\mathbb{E}\left\|\mathbf{u}_{y,t}\right\|^2 + \frac{2}{mK}\sigma_y^2.$$

*Proof.*

$$\mathbb{E}\|\left(\mathbf{u}_{x,t} - \mathbf{e}_{x,t}\right)\|^2 = \mathbb{E}\|\left(\mathbf{u}_{x,t} - \bar{\mathbf{e}}_{x,t}\right) + \left(\bar{\mathbf{e}}_{x,t} - \mathbf{e}_{x,t}\right)\|^2$$
$$\leq 2\mathbb{E}\|\left(\mathbf{u}_{x,t} - \bar{\mathbf{e}}_{x,t}\right)\|^2 + 2\mathbb{E}\|\left(\bar{\mathbf{e}}_{x,t} - \mathbf{e}_{x,t}\right)\|^2$$
$$\leq 4\mathbb{E}\|\bar{\mathbf{e}}_{x,t}\|^2 + 4\mathbb{E}\|\mathbf{u}_{x,t}\|^2 + \frac{2}{mK}\sigma_x^2,$$

where the second inequality follows from the fact that $\{\nabla_x f_i(\mathbf{z}_{t,i}^j, \xi_{t,i}^j) - \nabla_x f_i(\mathbf{z}_{t,i}^j)\}$ the martingale difference sequence (see Lemma 4 in [28]).

The bound of $\|\left(\mathbf{u}_{y,t} - \mathbf{e}_{y,t}\right)\|^2$ follows from the similar proof. $\qquad\square$

**Lemma 2** (One Round Progress for $\Phi$)**.**

$$\mathbb{E}\Phi(\mathbf{x}_{t+1}) - \Phi(\mathbf{x}_t) \leq -\frac{1}{2}\eta_x K \|\nabla\Phi(\mathbf{x}_t)\|^2 - \frac{1}{4}\eta_x K\|\nabla_x f(\mathbf{z}_t)\|^2 + 2L\eta_x^2 K^2 \mathbb{E}\|\mathbf{u}_{x,t}\|^2$$
$$+ \eta_x K \left(\frac{3}{2} + 2L\eta_x K\right) \mathbb{E}\|\bar{\mathbf{e}}_{x,t}\|^2 + \eta_x K \frac{L_f^2}{\mu^2}\|\nabla_y f(\mathbf{z}_t)\|^2 + \frac{L\eta_x^2 K}{m}\sigma_x^2.$$

*Proof.* Due to the $L$-smoothness of $\Phi(\mathbf{x})$, we have one step update in expectation conditioned on step $t$:

$$\mathbb{E}\Phi(\mathbf{x}_{t+1}) - \Phi(\mathbf{x}_t) \leq \langle\nabla\Phi(\mathbf{x}_t), \mathbb{E}[\mathbf{x}_{t+1} - \mathbf{x}_t]\rangle + \frac{L}{2}\mathbb{E}\|\mathbf{x}_{t+1} - \mathbf{x}_t\|^2$$
$$= \underbrace{\langle\nabla\Phi(\mathbf{x}_t), -\eta_x K\mathbb{E}[\mathbf{u}_{x,t} - \mathbf{e}_{x,t}]\rangle}_{A_1} + \underbrace{\frac{L}{2}\mathbb{E}\|\eta_x K(\mathbf{u}_{x,t} - \mathbf{e}_{x,t})\|^2}_{A_2}.$$

$$A_1 = \langle\nabla\Phi(\mathbf{x}_t), -\eta_x K\mathbb{E}(\nabla_x f(\mathbf{z}_t) - \bar{\mathbf{e}}_{x,t})\rangle$$
$$= -\frac{1}{2}\eta_x K\|\nabla\Phi(\mathbf{x}_t)\|^2 - \frac{1}{2}\eta_x K\mathbb{E}\|\nabla_x f(\mathbf{z}_t) - \bar{\mathbf{e}}_{x,t}\|^2 + \frac{1}{2}\eta_x K\mathbb{E}\|\nabla\Phi(\mathbf{x}_t) - \nabla_x f(\mathbf{z}_t) + \bar{\mathbf{e}}_{x,t}\|^2$$
$$\leq -\frac{1}{2}\eta_x K\|\nabla\Phi(\mathbf{x}_t)\|^2 - \frac{1}{4}\eta_x K\|\nabla_x f(\mathbf{z}_t)\|^2 + \frac{3}{2}\eta_x K\mathbb{E}\|\bar{\mathbf{e}}_{x,t}\|^2 + \eta_x K\|\nabla\Phi(\mathbf{x}_t) - \nabla_x f(\mathbf{z}_t)\|^2,$$

where the last inequality follows from $\|\mathbf{a} + \mathbf{b}\|^2 \geq \frac{1}{2}\|\mathbf{a}\|^2 - \|\mathbf{b}\|^2$ and $\|\mathbf{a} + \mathbf{b}\|^2 \leq 2\|\mathbf{a}\|^2 + 2\|\mathbf{b}\|^2$.

$$A_2 \leq 2L\eta_x^2 K^2\mathbb{E}\|\bar{\mathbf{e}}_{x,t}\|^2 + 2L\eta_x^2 K^2\mathbb{E}\|\mathbf{u}_{x,t}\|^2 + \frac{L\eta_x^2 K}{m}\sigma_x^2,$$

where the inequality is due to Lemma 1.

$$\|\nabla\Phi(\mathbf{x}_t) - \nabla_x f(\mathbf{z}_t)\|^2 = L_f^2\|\mathbf{y}(\mathbf{x}_t) - \mathbf{y}^*\|^2$$
$$\leq \frac{L_f^2}{\mu^2}\|\nabla_y f(\mathbf{z}_t)\|^2,$$

where the last inequality is due to the PL condition (Theorem 2 in [42]).

Combining pieces together, we have:

$$\mathbb{E}\Phi(\mathbf{x}_{t+1}) - \Phi(\mathbf{x}_t) = \underbrace{\langle\nabla\Phi(\mathbf{x}_t), -\eta_x K\mathbb{E}[\mathbf{u}_{x,t} - \mathbf{e}_{x,t}]\rangle}_{A_1} + \underbrace{\frac{L}{2}\mathbb{E}\|\eta_x K(\mathbf{u}_{x,t} - \mathbf{e}_{x,t})\|^2}_{A_2}$$
$$\leq -\frac{1}{2}\eta_x K\|\nabla\Phi(\mathbf{x}_t)\|^2 - \frac{1}{4}\eta_x K\|\nabla_x f(\mathbf{z}_t)\|^2 + 2L\eta_x^2 K^2\mathbb{E}\|\mathbf{u}_{x,t}\|^2$$
$$+ \eta_x K\left(\frac{3}{2} + 2L\eta_x K\right)\mathbb{E}\|\bar{\mathbf{e}}_{x,t}\|^2 + \eta_x K\frac{L_f^2}{\mu^2}\|\nabla_y f(\mathbf{z}_t)\|^2 + \frac{L\eta_x^2 K}{m}\sigma_x^2.$$

$\square$

**Lemma 3** (One Round Progress for $f$)**.**

$$f(\mathbf{z}_t) - \mathbb{E}f(\mathbf{z}_{t+1})$$
$$\leq \frac{3}{2}\eta_x K\|\nabla_x f(\mathbf{z}_t)\|^2 + 2L_f\eta_x^2 K^2\mathbb{E}\|\mathbf{u}_{x,t}\|^2 + \eta_x K\left(\frac{1}{2} + 2L_f\eta_x K\right)\mathbb{E}\|\bar{\mathbf{e}}_{x,t}\|^2 + \frac{L_f\eta_x^2 K}{m}\sigma_x^2$$
$$- \frac{1}{2}\eta_y K\|\nabla_y f(\mathbf{z}_t)\|^2 + 2L_f\eta_y^2 K^2\mathbb{E}\|\mathbf{u}_{y,t}\|^2 + \eta_y K\left(\frac{1}{2} + 2L_f\eta_y K\right)\mathbb{E}\|\bar{\mathbf{e}}_{y,t}\|^2 + \frac{L_f\eta_y^2 K}{m}\sigma_y^2.$$

*Proof.* Similarly, due to $L$-smoothness of $f(\mathbf{z})$, we have:

$$f(\mathbf{z}_t) - \mathbb{E}f(\mathbf{z}_{t+1}) \leq \eta_x K \mathbb{E} \langle \nabla_x f(\mathbf{z}_t), \mathbf{u}_{x,t} - \mathbf{e}_{x,t} \rangle - \eta_y K \mathbb{E} \langle \nabla_y f(\mathbf{z}_t), \mathbf{u}_{y,t} - \mathbf{e}_{y,t} \rangle$$

$$+ \frac{L_f \eta_x^2 K^2}{2} \mathbb{E} \|\mathbf{u}_{x,t} - \mathbf{e}_{x,t}\|^2 + \frac{L_f \eta_y^2 K^2}{2} \mathbb{E} \|\mathbf{u}_{y,t} - \mathbf{e}_{y,t}\|^2$$

$$= \eta_x K \mathbb{E} \langle \nabla_x f(\mathbf{z}_t), \nabla_x f(\mathbf{z}_t) - \bar{\mathbf{e}}_{x,t} \rangle - \eta_y K \mathbb{E} \langle \nabla_y f(\mathbf{z}_t), \nabla_y f(\mathbf{z}_t) - \bar{\mathbf{e}}_{y,t} \rangle$$

$$+ \frac{L_f \eta_x^2 K^2}{2} \mathbb{E} \|\mathbf{u}_{x,t} - \mathbf{e}_{x,t}\|^2 + \frac{L_f \eta_y^2 K^2}{2} \mathbb{E} \|\mathbf{u}_{y,t} - \mathbf{e}_{y,t}\|^2$$

$$\leq \frac{3}{2} \eta_x K \|\nabla_x f(\mathbf{z}_t)\|^2 + \frac{1}{2} \eta_x K \mathbb{E} \|\bar{\mathbf{e}}_{x,t}\|^2 - \frac{1}{2} \eta_y K \|\nabla_y f(\mathbf{z}_t)\|^2 + \frac{1}{2} \eta_y K \mathbb{E} \|\bar{\mathbf{e}}_{y,t}\|^2$$

$$+ \frac{L_f \eta_x^2 K^2}{2} \mathbb{E} \|\mathbf{u}_{x,t} - \mathbf{e}_{x,t}\|^2 + \frac{L_f \eta_y^2 K^2}{2} \mathbb{E} \|\mathbf{u}_{y,t} - \mathbf{e}_{y,t}\|^2$$

$$\leq \frac{3}{2} \eta_x K \|\nabla_x f(\mathbf{z}_t)\|^2 + 2 L_f \eta_x^2 K^2 \mathbb{E} \|\mathbf{u}_{x,t}\|^2 + \eta_x K \left( \frac{1}{2} + 2 L_f \eta_x K \right) \mathbb{E} \|\bar{\mathbf{e}}_{x,t}\|^2 + \frac{L_f \eta_x^2 K}{m} \sigma_x^2$$

$$- \frac{1}{2} \eta_y K \|\nabla_y f(\mathbf{z}_t)\|^2 + 2 L_f \eta_y^2 K^2 \mathbb{E} \|\mathbf{u}_{y,t}\|^2 + \eta_y K \left( \frac{1}{2} + 2 L_f \eta_y K \right) \mathbb{E} \|\bar{\mathbf{e}}_{y,t}\|^2 + \frac{L_f \eta_y^2 K}{m} \sigma_y^2.$$

$\square$

**Lemma 4** (Bounded Error for FSGDA).

$$\mathbb{E}\|\bar{\mathbf{e}}_{x,t}\|^2 \leq L_f^2 \bigg[ 40K^2 \eta_{x,l}^2 \|\nabla_x f(\mathbf{z}_t)\|^2 + 40K^2 \eta_{y,l}^2 \|\nabla_y f(\mathbf{z}_t)\|^2 + 40K^2 \eta_{x,l}^2 \sigma_{x,G}^2 + 40K^2 \eta_{y,l}^2 \sigma_{y,G}^2$$

$$+ 5K \eta_{x,l}^2 \sigma_x^2 + 5K \eta_{y,l}^2 \sigma_y^2 \bigg],$$

$$\mathbb{E}\|\bar{\mathbf{e}}_{y,t}\|^2 \leq L_f^2 \bigg[ 40K^2 \eta_{x,l}^2 \|\nabla_x f(\mathbf{z}_t)\|^2 + 40K^2 \eta_{y,l}^2 \|\nabla_y f(\mathbf{z}_t)\|^2 + 40K^2 \eta_{x,l}^2 \sigma_{x,G}^2 + 40K^2 \eta_{y,l}^2 \sigma_{y,G}^2$$

$$+ 5K \eta_{x,l}^2 \sigma_x^2 + 5K \eta_{y,l}^2 \sigma_y^2 \bigg].$$

*Proof.*

$$\mathbb{E}\|\bar{\mathbf{e}}_{x,t}\|^2 = \mathbb{E} \left\| \frac{1}{mK} \sum_{i \in S_t} \sum_{j \in [K]} \left( \nabla_x f_i(\mathbf{z}_t) - \nabla_x f_i(\mathbf{z}_{t,i}^j) \right) \right\|^2$$

$$\leq \mathbb{E} \left[ \frac{1}{K} \sum_{i \in S_t} \sum_{j \in [K]} \left\| \left( \nabla_x f_i(\mathbf{z}_t) - \nabla_x f_i(\mathbf{z}_{t,i}^j) \right) \right\|^2 \right]$$

$$\leq \frac{L_f^2}{MK} \sum_{i \in [M]} \sum_{j \in [K]} \mathbb{E} \left\| \left( \mathbf{z}_t - \mathbf{z}_{t,i}^j \right) \right\|^2$$

$$\mathbb{E} \left\| \left( \mathbf{z}_t - \mathbf{z}_{t,i}^{j+1} \right) \right\|^2 = \mathbb{E} \left[ \left\| \mathbf{x}_{t,i}^j - \mathbf{x}_t - \eta_{x,l} \nabla_x f_i(\mathbf{z}_{t,i}^j, \xi_{t,i}^j) \right\|^2 \right] + \mathbb{E} \left[ \left\| \mathbf{y}_{t,i}^j - \mathbf{y}_t - \eta_{y,l} \nabla_y f_i(\mathbf{z}_{t,i}^j, \xi_{t,i}^j) \right\|^2 \right]$$

$$\leq \mathbb{E} \left[ \left\| \mathbf{x}_{t,i}^j - \mathbf{x}_t - \eta_{x,l} \nabla_x f_i(\mathbf{z}_{t,i}^j) \right\|^2 \right] + \mathbb{E} \left[ \left\| \mathbf{y}_{t,i}^j - \mathbf{y}_t - \eta_{y,l} \nabla_y f_i(\mathbf{z}_{t,i}^j) \right\|^2 \right] + \eta_{x,l}^2 \sigma_x^2 + \eta_{y,l}^2 \sigma_y^2$$

$$\leq \left( 1 + \frac{1}{2K-1} \right) \left\| \mathbf{z}_{t,i}^j - \mathbf{z}_t \right\|^2 + 2K \eta_{x,l}^2 \left\| \nabla_x f_i \left( \mathbf{z}_{t,i}^j \right) \right\|^2 + 2K \eta_{y,l}^2 \left\| \nabla_y f_i \left( \mathbf{z}_{t,i}^j \right) \right\|^2 + \eta_{x,l}^2 \sigma_x^2 + \eta_{y,l}^2 \sigma_y^2$$

$$\leq \left( 1 + \frac{1}{2K-1} \right) \left\| \mathbf{z}_{t,i}^j - \mathbf{z}_t \right\|^2 + 4K \eta_{x,l}^2 \left\| \nabla_x f_i \left( \mathbf{z}_{t,i}^j \right) - \nabla_x f_i \left( \mathbf{z}_t \right) \right\|^2 + 4K \|\nabla_x f_i \left( \mathbf{z}_t \right)\|^2$$

$$+ 4K \eta_{y,l}^2 \left\| \nabla_y f_i \left( \mathbf{z}_{t,i}^j \right) - \nabla_y f_i \left( \mathbf{z}_t \right) \right\|^2 + 4K \eta_{y,l}^2 \|\nabla_y f_i \left( \mathbf{z}_t \right)\|^2 + \eta_{x,l}^2 \sigma_x^2 + \eta_{y,l}^2 \sigma_y^2$$

$$\leq \left(1 + \frac{1}{2K-1} + 4K \max\{L_f^2 \eta_{x,l}^2, L_f^2 \eta_{y,l}^2\}\right) \left\|\mathbf{z}_{t,i}^j - \mathbf{z}_t\right\|^2$$
$$+ 4K\eta_{x,l}^2 \left\|\nabla_x f_i(\mathbf{z}_t)\right\|^2 + 4K\eta_{y,l}^2 \left\|\nabla_y f_i(\mathbf{z}_t)\right\|^2 + \eta_{x,l}^2 \sigma_x^2 + \eta_{y,l}^2 \sigma_y^2$$
$$\leq \left(1 + \frac{1}{K-1}\right) \left\|\mathbf{z}_{t,i}^j - \mathbf{z}_t\right\|^2 + 4K \left\|\nabla_x f_i(\mathbf{z}_t)\right\|^2 + 4K\eta_{y,l}^2 \left\|\nabla_y f_i(\mathbf{z}_t)\right\|^2 + \eta_{x,l}^2 \sigma_x^2 + \eta_{y,l}^2 \sigma_y^2$$
$$\leq \sum_{\tau=0}^{j-1} \left(1 + \frac{1}{K-1}\right)^\tau \left[4K\eta_{x,l}^2 \left\|\nabla_x f_i(\mathbf{z}_t)\right\|^2 + 4K\eta_{y,l}^2 \left\|\nabla_y f_i(\mathbf{z}_t)\right\|^2 + \eta_{x,l}^2 \sigma_x^2 + \eta_{y,l}^2 \sigma_y^2\right]$$
$$\leq 20K^2 \eta_{x,l}^2 \left\|\nabla_x f_i(\mathbf{z}_t)\right\|^2 + 20K^2 \eta_{y,l}^2 \left\|\nabla_y f_i(\mathbf{z}_t)\right\|^2 + 5K\eta_{x,l}^2 \sigma_x^2 + 5K\eta_{y,l}^2 \sigma_y^2$$
$$\leq 40K^2 \eta_{x,l}^2 \left\|\nabla_x f(\mathbf{z}_t)\right\|^2 + 40K^2 \eta_{y,l}^2 \left\|\nabla_y f(\mathbf{z}_t)\right\|^2 + 40K^2 \eta_{x,l}^2 \sigma_{x,G}^2 + 40K^2 \eta_{y,l}^2 \sigma_{y,G}^2$$
$$+ 5K\eta_{x,l}^2 \sigma_x^2 + 5K\eta_{y,l}^2 \sigma_y^2,$$

where the first inequality is due to bounded variance of stochastic gradient, the second and third inequalities follow from the fact $\|\mathbf{a} + \mathbf{b}\|^2 \leq \left(1 + \frac{1}{\epsilon}\right)\|\mathbf{a}\|^2 + (1 + \epsilon)\|\mathbf{b}\|^2$, the forth inequality is due to smoothness of $f$ in $x$ and $y$, fifth inequality holds if

$$4K \max\{L_f^2 \eta_{x,l}^2, L_f^2 \eta_{y,l}^2\} \leq \frac{1}{2(K-1)(2K-1)}, \tag{7}$$

the second last inequality follows from the $\sum_{\tau=0}^{j-1} \left(1 + \frac{1}{K-1}\right)^\tau \leq (K-1)\left[\left(1 + \frac{1}{K-1}\right)^K - 1\right] \leq 5K$, and the last inequality is due to the Assumption 4.

Plugging into the bound of $\|\bar{\mathbf{e}}_{x,t}\|^2$, we have:

$$\|\bar{\mathbf{e}}_{x,t}\|^2 \leq L_f^2 \left[40K^2 \eta_{x,l}^2 \left\|\nabla_x f(\mathbf{z}_t)\right\|^2 + 40K^2 \eta_{y,l}^2 \left\|\nabla_y f(\mathbf{z}_t)\right\|^2 + 40K^2 \eta_{x,l}^2 \sigma_{x,G}^2 + 40K^2 \eta_{y,l}^2 \sigma_{y,G}^2\right.$$
$$\left. + 5K\eta_{x,l}^2 \sigma_x^2 + 5K\eta_{y,l}^2 \sigma_y^2\right].$$

The bound of $\|\bar{\mathbf{e}}_{y,t}\|^2$ follows from the similar proof.

$\square$

**Theorem 2** (Convergence Rate for FSGDA). *Under Assumptions 1- 4, define $\mathcal{L}_t = \Phi(\mathbf{x}_t) - \frac{1}{10}f(\mathbf{x}_t, \mathbf{y}_t)$, if the learning rates $\eta_{x,g}$, $\eta_{x,l}$, $\eta_{y,g}$, and $\eta_{y,l}$ satisfy:*

$$8K(K-1)(2K-1)L_f^2 \max\{\eta_{x,l}^2, \eta_{y,l}^2\} \leq 1,$$
$$a_1 - a_3 40 L_f^2 K^2 \eta_{x,l}^2 - \frac{\eta_y}{\eta_x} a_4 40 L_f^2 K^2 \eta_{x,l}^2 \geq 0,$$
$$a_2 - a_3 \frac{\eta_x}{\eta_y} 40 L_f^2 K^2 \eta_{y,l}^2 - a_4 40 L_f^2 K^2 \eta_{y,l}^2 \geq 0,$$

*where $a_1 = \left(\frac{1}{10} - 2(2L + \frac{1}{5}L_f)\eta_x K\right), a_2 = \left(\frac{1}{20} - \frac{2}{5}L_f \eta_y K - \frac{\eta_x}{\eta_y}\frac{L_f^2}{\mu^2}\right), a_3 = \left(\frac{31}{20} + (2L + \frac{1}{5}L_f)\eta_x K\right)$ and $a_4 = \left(\frac{1}{20} + \frac{1}{5}L_f \eta_y K\right)$, then the output sequence $\{\mathbf{x}_t\}$ generated by FSGDA satisfies:*

$$\frac{1}{T}\sum_{t=0}^{T-1} \mathbb{E}\|\nabla\Phi(\mathbf{x}_t)\|^2 \leq \underbrace{\frac{2(\mathcal{L}_0 - \mathcal{L}_T)}{\eta_x K T}}_{\text{optimization error}} + \underbrace{\frac{2\eta_x}{m}\left(L + \frac{L_f}{100}\right)\sigma_x^2 + \frac{L_f \eta_y^2}{5m\eta_x}\sigma_y^2}_{\text{statistical error}} + \underbrace{\psi_3}_{\substack{\text{local} \\ \text{update error}}} + \underbrace{\psi_4}_{\substack{\text{sampling} \\ \text{variance}}}.$$

*Here, $\psi_3$ and $\psi_4$ are defined as follows:*

$$\psi_3 = 2\left(a_3 L_f^2 + a_4 \frac{\eta_y}{\eta_x} L_f^2\right)\left[40K^2 \eta_{x,l}^2 \sigma_{x,G}^2 + 40K^2 \eta_{y,l}^2 \sigma_{y,G}^2 + 5K\eta_{x,l}^2 \sigma_x^2 + 5K\eta_{y,l}^2 \sigma_y^2\right],$$
$$\psi_4 = \left((2L + \frac{1}{5}L_f)\eta_x K\right)\left(1 - \frac{m}{M}\right)\frac{2}{m}\sigma_{x,G}^2 + \frac{2}{5m}L_f \eta_y K \frac{\eta_y}{\eta_x}\left(1 - \frac{m}{M}\right)\sigma_{y,G}^2,$$

*Proof.* Define potential function $\mathcal{L}_t = \Phi(\mathbf{x}_t) - \frac{1}{10}f(\mathbf{z}_t)$,

$$\mathbb{E}L_{t+1} - \mathcal{L}_t = \mathbb{E}\Phi(\mathbf{x}_{t+1}) - \Phi(\mathbf{x}_t) + \frac{1}{10}\left(f(\mathbf{z}_t) - \mathbb{E}f(\mathbf{z}_{t+1})\right)$$

$$\leq -\frac{1}{2}\eta_x K\|\nabla\Phi(\mathbf{x}_t)\|^2 - \frac{1}{10}\eta_x K\|\nabla_x f(\mathbf{z}_t)\|^2 + \left((2L + \frac{1}{5}L_f)\eta_x^2 K^2\right)\mathbb{E}\|\mathbf{u}_{x,t}\|^2$$

$$- \eta_y K\left(\frac{1}{20} - \frac{\eta_x}{\eta_y}\frac{L_f^2}{\mu^2}\right)\|\nabla_y f(\mathbf{z}_t)\|^2 + \frac{1}{5}L_f\eta_y^2 K^2\mathbb{E}\|\mathbf{u}_{y,t}\|^2$$

$$+ \eta_x K\left(\frac{31}{20} + (2L + \frac{1}{5}L_f)\eta_x K\right)\mathbb{E}\|\bar{\mathbf{e}}_{x,t}\|^2 + \eta_y K\left(\frac{1}{20} + \frac{1}{5}L_f\eta_y K\right)\mathbb{E}\|\bar{\mathbf{e}}_{y,t}\|^2$$

$$+ \frac{\eta_x^2 K}{m}\left(L + \frac{L_f}{10}\right)\sigma_x^2 + \frac{L_f\eta_y^2 K}{10m}\sigma_y^2$$

$$\leq -\frac{1}{2}\eta_x K\|\nabla\Phi(\mathbf{x}_t)\|^2 - \eta_x K\underbrace{\left(\frac{1}{10} - 2(2L + \frac{1}{5}L_f)\eta_x K\right)}_{a_1}\|\nabla_x f(\mathbf{z}_t)\|^2$$

$$- \eta_y K\underbrace{\left(\frac{1}{20} - \frac{2}{5}L_f\eta_y K - \frac{\eta_x}{\eta_y}\frac{L_f^2}{\mu^2}\right)}_{a_2}\|\nabla_y f(\mathbf{z}_t)\|^2$$

$$+ \left((2L + \frac{1}{5}L_f)\eta_x^2 K^2\right)\left(1 - \frac{m}{M}\right)\frac{2}{m}\sigma_{x,G}^2 + \frac{1}{5}L_f\eta_y^2 K^2\left(1 - \frac{m}{M}\right)\frac{2}{m}\sigma_{y,G}^2$$

$$+ \eta_x K\underbrace{\left(\frac{31}{20} + (2L + \frac{1}{5}L_f)\eta_x K\right)}_{a_3}\mathbb{E}\|\bar{\mathbf{e}}_{x,t}\|^2 + \eta_y K\underbrace{\left(\frac{1}{20} + \frac{1}{5}L_f\eta_y K\right)}_{a_4}\mathbb{E}\|\bar{\mathbf{e}}_{y,t}\|^2$$

$$+ \frac{\eta_x^2 K}{m}\left(L + \frac{L_f}{10}\right)\sigma_x^2 + \frac{L_f\eta_y^2 K}{10m}\sigma_y^2$$

$$\leq -\frac{1}{2}\eta_x K\|\nabla\Phi(\mathbf{x}_t)\|^2 + \frac{\eta_x^2 K}{m}\left(L + \frac{L_f}{10}\right)\sigma_x^2 + \frac{L_f\eta_y^2 K}{10m}\sigma_y^2$$

$$+ \left((2L + \frac{1}{5}L_f)\eta_x^2 K^2\right)\left(1 - \frac{m}{M}\right)\frac{2}{m}\sigma_{x,G}^2 + \frac{1}{5}L_f\eta_y^2 K^2\left(1 - \frac{m}{M}\right)\frac{2}{m}\sigma_{y,G}^2$$

$$+ K\left(a_3 L_f^2\eta_x + a_4\eta_y L_f^2\right)\left[40K^2\eta_{x,l}^2\sigma_{x,G}^2 + 40K^2\eta_{y,l}^2\sigma_{y,G}^2 + 5K\eta_{x,l}^2\sigma_x^2 + 5K\eta_{y,l}^2\sigma_y^2\right],$$

where the second inequality is due to $\mathbb{E}\|\mathbf{u}_{x,t}\|^2 \leq 2\|\nabla_x f(\mathbf{z}_t)\|^2 + 2\left(1 - \frac{m}{M}\right)\frac{\sigma_{x,G}^2}{m}$ and $\mathbb{E}\|\mathbf{u}_{y,t}\|^2 \leq 2\|\nabla_y f(\mathbf{z}_t)\|^2 + 2\left(1 - \frac{m}{M}\right)\frac{\sigma_{y,G}^2}{m}$, the last inequality follows from the conditions:

$$a_1 - a_3 40L_f^2 K^2\eta_{x,l}^2 - \frac{\eta_y}{\eta_x}a_4 40L_f^2 K^2\eta_{x,l}^2 \geq 0, \tag{8}$$

$$a_2 - a_3\frac{\eta_x}{\eta_y}40L_f^2 K^2\eta_{y,l}^2 - a_4 40L_f^2 K^2\eta_{y,l}^2 \geq 0. \tag{9}$$

Telescoping and rearranging, we have:

$$\frac{1}{T}\sum_{t=0}^{T-1}\|\nabla\Phi(\mathbf{x}_t)\|^2 \leq \frac{2(\mathcal{L}_0 - \mathcal{L}_*)}{\eta_x KT} + \frac{2\eta_x}{m}\left(L + \frac{L_f}{100}\right)\sigma_x^2 + \frac{L_f\eta_y^2}{5m\eta_x}\sigma_y^2$$

$$+ \left((2L + \frac{1}{5}L_f)\eta_x K\right)\left(1 - \frac{m}{M}\right)\frac{2}{m}\sigma_{x,G}^2 + \frac{1}{5}L_f\eta_y K\frac{\eta_y}{\eta_x}\left(1 - \frac{m}{M}\right)\frac{2}{m}\sigma_{y,G}^2$$

$$+ 2\left(a_3 L_f^2 + a_4\frac{\eta_y}{\eta_x}L_f^2\right)\left[40K^2\eta_{x,l}^2\sigma_{x,G}^2 + 40K^2\eta_{y,l}^2\sigma_{y,G}^2 + 5K\eta_{x,l}^2\sigma_x^2 + 5K\eta_{y,l}^2\sigma_y^2\right].$$

$\square$

## B.2 Proof for SAGDA **Option I**

For SAGDA Option I, the update rule is:

$$\mathbf{x}_{t+1} = \mathbf{x}_t - \eta_{x,g}\eta_{x,l} \left[ \frac{1}{m} \sum_{i \in S_t} \sum_{j \in [K]} \left( \nabla_x f_i(\mathbf{z}_{t,i}^j, \xi_{t,i}^j) - \mathbf{v}_x^i + \bar{\mathbf{v}}_{x,t} \right) \right],$$

$$\mathbf{y}_{t+1} = \mathbf{y}_t + \eta_{y,g}\eta_{y,l} \left[ \frac{1}{m} \sum_{i \in S_t} \sum_{j \in [K]} \left( \nabla_y f_i(\mathbf{z}_{t,i}^j, \xi_{t,i}^j) - \mathbf{v}_y^i + \bar{\mathbf{v}}_{y,t} \right) \right],$$

$$\mathbf{e}_{x,t} = \frac{1}{mK} \sum_{i \in S_t} \sum_{j \in [K]} \left[ \nabla_x f_i(\mathbf{z}_t) - \left( \nabla_x f_i(\mathbf{z}_{t,i}^j, \xi_{t,i}^j) - \mathbf{v}_x^i + \bar{\mathbf{v}}_{x,t} \right) \right]$$

$$\bar{\mathbf{e}}_{x,t} = \mathbb{E}[\mathbf{e}_{x,t}] = \frac{1}{mK} \sum_{i \in S_t} \sum_{j \in [K]} \left( \nabla_x f_i(\mathbf{z}_t) - \nabla_x f_i(\mathbf{z}_{t,i}^j) \right),$$

$$\mathbf{e}_{y,t} = \frac{1}{mK} \sum_{i \in S_t} \sum_{j \in [K]} \left[ \nabla_y f_i(\mathbf{z}_t) - \left( \nabla_y f_i(\mathbf{z}_{t,i}^j, \xi_{t,i}^j) - \mathbf{v}_y^i + \bar{\mathbf{v}}_{y,t} \right) \right]$$

$$\bar{\mathbf{e}}_{y,t} = \mathbb{E}[\mathbf{e}_{y,t}] = \frac{1}{mK} \sum_{i \in S_t} \sum_{j \in [K]} \left( \nabla_y f_i(\mathbf{z}_t) - \nabla_y f_i(\mathbf{z}_{t,i}^j) \right),$$

where we define $\mathbf{v}_x^i = \nabla_x f_i(\mathbf{w}_{t,i}, \xi)$ and $\bar{\mathbf{v}}_{x,t} = \frac{1}{M} \sum_{i \in [M]} \mathbf{v}_x^i$ with a sequence of parameters $\mathbf{w}_{t,i}$ such that

$$\mathbf{w}_{t,i} := \begin{cases} \mathbf{z}_{t-1}, & \text{if } i \in S_{t-1}, \\ \mathbf{w}_{t-1,i}, & \text{otherwise.} \end{cases}$$

We further have the following definition for notational clarity:

$$\Delta\mathbf{x}_t = \frac{1}{mK} \sum_{i \in S_t} \sum_{j \in [K]} \left[ \nabla_x f_i(\mathbf{z}_{t,i}^j, \xi_{t,i}^j) - \mathbf{v}_x^i + \bar{\mathbf{v}}_{x,t} \right],$$

$$\Delta\mathbf{y}_t = \frac{1}{mK} \sum_{i \in S_t} \sum_{j \in [K]} \left[ \nabla_y f_i(\mathbf{z}_{t,i}^j, \xi_{t,i}^j) - \mathbf{v}_y^i + \bar{\mathbf{v}}_{y,t} \right],$$

$$\Psi_t = \frac{1}{MK} \sum_{i \in [M]} \sum_{j \in [K]} \mathbb{E} \left\| \mathbf{z}_{t,i}^j - \mathbf{z}_t \right\|^2,$$

$$\Gamma_t = \frac{1}{M} \sum_{i \in [M]} \mathbb{E} \left\| \mathbf{w}_{t,i} - \mathbf{z}_t \right\|^2.$$

**Lemma 5** (Iterative Control Variate)**.**

$$\Gamma_t = \left( 1 - \frac{m}{2M} \right) \Gamma_{t-1} + \left( \frac{m}{M} + \frac{M}{m} - 1 \right) \mathbb{E} \left\| \mathbf{z}_t - \mathbf{z}_{t-1} \right\|^2.$$

*Proof.*

$$\Gamma_t = \frac{1}{M} \sum_{i \in [M]} \mathbb{E} \left\| \mathbf{w}_{t,i} - \mathbf{z}_t \right\|^2$$

$$= \left( 1 - \frac{m}{M} \right) \frac{1}{M} \sum_{i \in [M]} \mathbb{E} \left\| \mathbf{w}_{t-1,i} - \mathbf{z}_t \right\|^2 + \frac{m}{M} \mathbb{E} \left\| \mathbf{z}_{t-1} - \mathbf{z}_t \right\|^2$$

$$\leq \left( 1 - \frac{m}{M} \right) \left( 1 + \frac{1}{b} \right) \Gamma_{t-1} + \left[ \left( 1 - \frac{m}{M} \right) (1 + b) + \frac{m}{M} \right] \mathbb{E} \left\| \mathbf{z}_t - \mathbf{z}_{t-1} \right\|^2$$

$$= \left( 1 - \frac{m}{2M} \right) \Gamma_{t-1} + \left( \frac{m}{M} + \frac{M}{m} - 1 \right) \mathbb{E} \left\| \mathbf{z}_t - \mathbf{z}_{t-1} \right\|^2,$$

where we set $b = \frac{2M}{m} - 1$. $\square$

**Lemma 6** (Local Step Distance for SAGDA Option I). $\forall i \in [M], j \in [K]$, *we can bound the local step distance as follows:*

$$\frac{1}{M} \sum_{i \in [M]} \mathbb{E} \left\| \left( \mathbf{z}_{t,i}^j - \mathbf{z}_t \right) \right\|^2 \leq 160K^2 \left( \eta_{x,l}^2 + \eta_{y,l}^2 \right) L_f^2 \Gamma_t + 10K^2 \left( \eta_{x,l}^2 \sigma_x^2 + \eta_{y,l}^2 \sigma_y^2 \right)$$

$$+ 40K^2 \left( \eta_{x,l}^2 \mathbb{E} \left\| \nabla_x f(\mathbf{z}_t) \right\|^2 + \eta_{y,l}^2 \mathbb{E} \left\| \nabla_y f(\mathbf{z}_t) \right\|^2 \right).$$

*Proof.* First, we bound the local update as follows:

$$\mathbb{E} \left\| \left( \nabla_x f_i(\mathbf{z}_{t,i}^j, \xi_{t,i}^j) - \mathbf{v}_x^i + \bar{\mathbf{v}}_{x,t} \right) \right\|^2$$

$$\leq 4 \left[ \mathbb{E} \left\| \nabla_x f_i(\mathbf{z}_{t,i}^j) - \nabla_x f_i(\mathbf{z}_t) \right\|^2 + \mathbb{E} \left\| \mathbb{E}[\mathbf{v}_x^i] - \nabla_x f_i(\mathbf{z}_t) \right\|^2 + \mathbb{E} \left\| \mathbb{E}[\bar{\mathbf{v}}_{x,t}] - \nabla_x f(\mathbf{z}_t) \right\|^2 \right.$$

$$\left. + \left\| \nabla_x f(\mathbf{z}_t) \right\|^2 \right] + \sigma_x^2$$

$$\leq 4L_f^2 \mathbb{E} \left\| \mathbf{z}_{t,i}^j - \mathbf{z}_t \right\|^2 + 4L_f^2 \mathbb{E} \left\| \mathbf{w}_{t,i} - \mathbf{z}_t \right\|^2 + 4L_f^2 \mathbb{E} \left\| \mathbb{E}[\bar{\mathbf{v}}_{x,t}] - \nabla_x f(\mathbf{z}_t) \right\|^2$$

$$+ 4\mathbb{E} \left\| \nabla_x f(\mathbf{z}_t) \right\|^2 + \sigma_x^2.$$

That is,

$$\frac{1}{M} \sum_{i \in [M]} \mathbb{E} \left\| \left( \nabla_x f_i(\mathbf{z}_{t,i}^j, \xi_{t,i}^j) - \mathbf{v}_x^i + \bar{\mathbf{v}}_{x,t} \right) \right\|^2$$

$$\leq 4L_f^2 \frac{1}{M} \sum_{i \in [M]} \mathbb{E} \left\| \mathbf{z}_{t,i}^j - \mathbf{z}_t \right\|^2 + 8L_f^2 \Gamma_t + \sigma_x^2 + 4\mathbb{E} \left\| \nabla_x f(\mathbf{z}_t) \right\|^2.$$

$$\frac{1}{M} \sum_{i \in [M]} \mathbb{E} \left[ \left\| \mathbf{x}_{t,i}^{j+1} - \mathbf{x}_t \right\|^2 \right] = \frac{1}{M} \sum_{i \in [M]} \mathbb{E} \left[ \left\| \mathbf{x}_{t,i}^j - \mathbf{x}_t - \eta_{x,l} \left( \nabla_x f_i(\mathbf{z}_{t,i}^j, \xi_{t,i}^j) - \mathbf{v}_{x,t}^i + \bar{\mathbf{v}}_{x,t} \right) \right\|^2 \right]$$

$$\leq \left( 1 + \frac{1}{2K-1} \right) \frac{1}{M} \sum_{i \in [M]} \mathbb{E} \left\| \mathbf{x}_{t,i}^j - \mathbf{x}_t \right\|^2 + 2K \eta_{x,l}^2 \frac{1}{M} \sum_{i \in [M]} \mathbb{E} \left\| \nabla_x f_i(\mathbf{z}_{t,i}^j, \xi_{t,i}^j) - \mathbf{v}_{x,t}^i + \bar{\mathbf{v}}_{x,t} \right\|^2$$

$$\leq \left( 1 + \frac{1}{2K-1} + 8KL_f^2 \eta_{x,l}^2 \right) \frac{1}{M} \sum_{i \in [M]} \mathbb{E} \left\| \mathbf{x}_{t,i}^j - \mathbf{x}_t \right\|^2 + 32K \eta_{x,l}^2 L_f^2 \Gamma_t$$

$$+ 2K \eta_{x,l}^2 \sigma_x^2 + 8K \eta_{x,l}^2 \mathbb{E} \left\| \nabla_x f(\mathbf{z}_t) \right\|^2.$$

We can bound $\left\| \mathbf{y}_{t,i}^{j+1} - \mathbf{y}_t \right\|^2$ in the same way, and then we have

$$\frac{1}{M} \sum_{i \in [M]} \mathbb{E} \left\| \left( \mathbf{z}_{t,i}^{j+1} - \mathbf{z}_t \right) \right\|^2$$

$$\leq \left( 1 + \frac{1}{2K-1} + 8KL_f^2 \max\{\eta_{x,l}^2, \eta_{y,l}^2\} \right) \frac{1}{M} \sum_{i \in [M]} \mathbb{E} \left\| \mathbf{z}_{t,i}^j - \mathbf{z}_t \right\|^2 + \left[ 32K \left( \eta_{x,l}^2 + \eta_{y,l}^2 \right) L_f^2 \Gamma_t \right.$$

$$\left. + 2K \left( \eta_{x,l}^2 \sigma_x^2 + \eta_{y,l}^2 \sigma_y^2 \right) + 8K \left( \eta_{x,l}^2 \mathbb{E} \left\| \nabla_x f(\mathbf{z}_t) \right\|^2 + \eta_{y,l}^2 \mathbb{E} \left\| \nabla_y f(\mathbf{z}_t) \right\|^2 \right) \right]$$

$$\leq \left( 1 + \frac{1}{K-1} \right) \frac{1}{M} \sum_{i \in [M]} \mathbb{E} \left\| \mathbf{z}_{t,i}^j - \mathbf{z}_t \right\|^2 \left[ 32K \left( \eta_{x,l}^2 + \eta_{y,l}^2 \right) L_f^2 \Gamma_t \right.$$

$$\left. + 2K \left( \eta_{x,l}^2 \sigma_x^2 + \eta_{y,l}^2 \sigma_y^2 \right) + 8K \left( \eta_{x,l}^2 \mathbb{E} \left\| \nabla_x f(\mathbf{z}_t) \right\|^2 + \eta_{y,l}^2 \mathbb{E} \left\| \nabla_y f(\mathbf{z}_t) \right\|^2 \right) \right]$$

$$\leq \sum_{\tau=0}^{j-1} \left(1 + \frac{1}{K-1}\right)^{\tau} \left[32K \left(\eta_{x,l}^2 + \eta_{y,l}^2\right) L_f^2 \Gamma_t \right.$$

$$+ 2K \left(\eta_{x,l}^2 \sigma_x^2 + \eta_{y,l}^2 \sigma_y^2\right) + 8K \left(\eta_{x,l}^2 \mathbb{E} \left\|\nabla_x f(\mathbf{z}_t)\right\|^2 + \eta_{y,l}^2 \mathbb{E} \left\|\nabla_y f(\mathbf{z}_t)\right\|^2\right) \right]$$

$$\leq 160K^2 \left(\eta_{x,l}^2 + \eta_{y,l}^2\right) L_f^2 \Gamma_t + 10K^2 \left(\eta_{x,l}^2 \sigma_x^2 + \eta_{y,l}^2 \sigma_y^2\right)$$

$$+ 40K^2 \left(\eta_{x,l}^2 \mathbb{E} \left\|\nabla_x f(\mathbf{z}_t)\right\|^2 + \eta_{y,l}^2 \mathbb{E} \left\|\nabla_y f(\mathbf{z}_t)\right\|^2\right).$$

The learning rates should satisfy

$$4K \max\{L_f^2 \eta_{x,l}^2, L_f^2 \eta_{y,l}^2\} \leq \frac{1}{2(K-1)(2K-1)}, \tag{10}$$

$\square$

**Lemma 7.**

$$\mathbb{E}\|\Delta \mathbf{x}_t\|^2 \leq 4L_f^2 \Psi_t + 4L_f^2 \Gamma_t + 4 \left\|\nabla_x f(\mathbf{z}_t)\right\|^2 + \frac{9}{mK}\sigma_x^2,$$

$$\mathbb{E}\|\Delta \mathbf{y}_t\|^2 \leq 4L_f^2 \Psi_t + 4L_f^2 \Gamma_t + 4 \left\|\nabla_y f(\mathbf{z}_t)\right\|^2 + \frac{9}{mK}\sigma_y^2,$$

$$\mathbb{E}\|\mathbf{z}_{t+1} - \mathbf{z}_t\|^2 \leq 4L_f^2 K^2 \left(\eta_x^2 + \eta_y^2\right) \Psi_t + 4L_f^2 K^2 \left(\eta_x^2 + \eta_y^2\right) \Gamma_t$$

$$+ 4K^2 \left(\eta_x^2 \left\|\nabla_x f(\mathbf{z}_t)\right\|^2 + \eta_y^2 \left\|\nabla_y f(\mathbf{z}_t)\right\|^2\right) + \frac{9K}{m} \left(\eta_x^2 \sigma_x^2 + \eta_y^2 \sigma_y^2\right).$$

*Proof.*

$$\mathbb{E}\|\Delta \mathbf{x}_t\|^2 \leq \mathbb{E} \left\| \frac{1}{mK} \sum_{i \in S_t} \sum_{j \in [K]} \left[\nabla_x f_i(\mathbf{z}_{t,i}^j) - \mathbb{E}[\mathbf{v}_x^i] + \mathbb{E}[\bar{\mathbf{v}}_{x,t}]\right] \right\|^2 + \frac{9}{mK}\sigma_x^2$$

$$\leq \frac{4}{MK} \sum_{i \in [M]} \sum_{j \in [K]} \left[\mathbb{E} \left\|\nabla_x f_i(\mathbf{z}_{t,i}^j) - \nabla_x f_i(\mathbf{z}_t)\right\|^2 + \mathbb{E} \left\|\mathbb{E}[\mathbf{v}_x^i] - \nabla_x f_i(\mathbf{z}_t)\right\|^2 \right.$$

$$+ \mathbb{E} \left\|\mathbb{E}[\bar{\mathbf{v}}_{x,t}] - \nabla_x f(\mathbf{z}_t)\right\|^2 + \left\|\nabla_x f(\mathbf{z}_t)\right\|^2 \right] + \frac{9}{mK}\sigma_x^2$$

$$\leq \frac{4}{MK} \sum_{i \in [M]} \sum_{j \in [K]} \left[L_f^2 \mathbb{E} \left\|\mathbf{z}_{t,i}^j - \mathbf{z}_t\right\|^2 + L_f^2 \mathbb{E} \left\|\mathbf{w}_{t,i} - \mathbf{z}_t\right\|^2 + \left\|\nabla_x f(\mathbf{z}_t)\right\|^2 \right] + \frac{9}{mK}\sigma_x^2$$

$$= 4L_f^2 \Psi_t + 4L_f^2 \Gamma_t + 4 \left\|\nabla_x f(\mathbf{z}_t)\right\|^2 + \frac{9}{mK}\sigma_x^2,$$

$\mathbb{E}[\mathbf{v}_{x,t}^i] = \nabla_x f_i(\mathbf{z}_t)$ and $\mathbb{E}[\bar{\mathbf{v}}_{x,t}] = \nabla_x f(\mathbf{z}_t)$ where the second inequality is due to Lemma 4 in [28]).

The bound of $\|(\mathbf{u}_{y,t} - \mathbf{e}_{y,t})\|^2$ follows from the similar proof. $\square$

**Lemma 8** (Bounded Error for SAGDA Option I).

$$\mathbb{E}\|\bar{\mathbf{e}}_{x,t}\|^2 \leq L_f^2 \Psi_t,$$

$$\mathbb{E}\|\bar{\mathbf{e}}_{y,t}\|^2 \leq L_f^2 \Psi_t.$$

*Proof.*

$$\mathbb{E}\|\bar{\mathbf{e}}_{x,t}\|^2 = \mathbb{E} \left\| \frac{1}{mK} \sum_{i \in S_t} \sum_{j \in [K]} \left(\nabla_x f_i(\mathbf{z}_t) - \nabla_x f_i(\mathbf{z}_{t,i}^j)\right) \right\|^2$$

$$\leq \frac{1}{mK} \mathbb{E} \left[ \sum_{i \in S_t} \sum_{j \in [K]} \left\|\left(\nabla_x f_i(\mathbf{z}_t) - \nabla_x f_i(\mathbf{z}_{t,i}^j)\right)\right\|^2 \right]$$

$$\leq \frac{L_f^2}{MK} \sum_{i\in[M],j\in[K]} \mathbb{E}\left\|\left(\mathbf{z}_t - \mathbf{z}_{t,i}^j\right)\right\|^2$$

$$= L_f^2 \Psi_t.$$

$\mathbb{E}\|\bar{\mathbf{e}}_{y,t}\|^2$ has the same bounds. $\qquad\square$

**Theorem 1** (Convergence Rate of SAGDA). *Under Assumptions 1- 3, define* $\mathcal{L}_t = \Phi(\mathbf{x}_t) - \frac{1}{10}f(\mathbf{x}_t, \mathbf{y}_t)$, *the output sequence* $\{\mathbf{x}_t\}$ *generated by* SAGDA *satisfies:*

• *For Option I with learning rates* $\eta_{x,g}$, $\eta_{x,l}$, $\eta_{y,g}$, *and* $\eta_{y,l}$ *satisfying*

$$8K(K-1)(2K-1)L_f^2 \max\{\eta_{x,l}^2, \eta_{y,l}^2\} \leq 1,$$

$$\frac{1}{2} - 4a_2 L_f^2 K^2 \left(\eta_x^2 + \eta_y^2\right) - \left(a_1 + a_2 4L_f^2 K^2 \left(\eta_x^2 + \eta_y^2\right)\right) 160K^2 \left(\eta_{x,l}^2 + \eta_{y,l}^2\right) L_f^2 \geq 0,$$

$$\left[\frac{1}{10}\eta_x K - 4a_2 K^2 \eta_x^2\right] - \left[a_1 + a_2 4L_f^2 K^2 \left(\eta_x^2 + \eta_y^2\right)\right] 40K^2 \eta_{x,l}^2 \geq 0,$$

$$\left[\eta_y K \left(\frac{1}{20} - \frac{\eta_x}{\eta_y}\frac{L_f^2}{\mu^2}\right) - 4a_2 K^2 \eta_y^2\right] - \left[a_1 + a_2 4L_f^2 K^2 \left(\eta_x^2 + \eta_y^2\right)\right] 40K^2 \eta_{y,l}^2 \geq 0,$$

*where* $a_1 = KL_f^2\left(\frac{31}{20}\eta_x + \frac{1}{20}\eta_y\right)$ *and* $a_2 = \frac{1}{2}\left(L + \frac{L_f}{10}\right) + 1 + \frac{M^2}{m^2} - \frac{M}{m}$, *it holds that*

$$\frac{1}{T}\sum_{t=0}^{T-1}\mathbb{E}\|\nabla\Phi(\mathbf{x}_t)\|^2 \leq \underbrace{\frac{2\left(\mathcal{L}_0 - \mathcal{L}_*\right)}{\eta_x KT}}_{\text{optimization error}} + \underbrace{\left[\left(L + \frac{L_f}{10}\right) + 4\right]\frac{9}{m\eta_x}\left(\eta_x^2\sigma_x^2 + \eta_y^2\sigma_y^2\right)}_{\text{statistical error}} + \underbrace{\psi_1}_{\text{local update error}}$$

*where* $\psi_1$ *is defined as follows:*

$$\psi_1 = \left[L_f^2\left(\frac{31}{20} + \frac{1}{20}\frac{\eta_y}{\eta_x}\right) + \left[\frac{1}{2}\left(L + \frac{L_f}{10}\right) + 2\right]4L_f^2 K\left(\eta_x + \frac{\eta_y^2}{\eta_x}\right)\right]\left[20K^2\left(\eta_{x,l}^2\sigma_x^2 + \eta_{y,l}^2\sigma_y^2\right)\right].$$

• *For Option II with learning rates* $\eta_{x,g}$, $\eta_{x,l}$, $\eta_{y,g}$, *and* $\eta_{y,l}$ *satisfying*

$$8K(K-1)(2K-1)L_f^2 \max\{\eta_{x,l}^2, \eta_{y,l}^2\} \leq 1,$$

$$\frac{1}{10}\eta_x K - \left(2\left(L + \frac{L_f}{10}\right)\eta_x^2 K^2 + 40K^2\eta_{x,l}^2 b_1\right) \geq 0,$$

$$\eta_y K\left(\frac{1}{20} - \frac{\eta_x}{\eta_y}\frac{L_f^2}{\mu^2}\right) - \left(\frac{1}{5}L_f\eta_y^2 K^2 + 40K^2\eta_{y,l}^2 b_1\right) \geq 0,$$

*where* $b_1 = L_f^2\left[\frac{31}{20}\eta_x K + \frac{1}{20}\eta_y K + 2\left(L + \frac{L_f}{10}\right)\eta_x^2 K^2 + \frac{1}{5}L_f\eta_y^2 K^2\right]$, *it holds that*

$$\frac{1}{T}\sum_{t=0}^{T-1}\mathbb{E}\|\nabla\Phi(\mathbf{x}_t)\|^2 \leq \frac{2\left(\mathcal{L}_0 - \mathcal{L}_*\right)}{\eta_x KT} + \left[\left(L + \frac{L_f}{10}\right)\frac{9\eta_x}{m}\sigma_x^2 + +\frac{9}{10}L_f\frac{\eta_y^2}{m\eta_x}\sigma_y^2\right] + \psi_2.$$

*where* $\psi_2$ *is defined as follows:*

$$\psi_2 = L_f^2\left[\frac{31}{20}K + \frac{1}{20}\frac{\eta_y}{\eta_x}K + 2\left(L + \frac{L_f}{10}\right)\eta_x K^2 + \frac{1}{5}L_f\frac{\eta_y^2}{\eta_x}K^2\right]\left[10\left(16K+1\right)\right]\left(\eta_{x,l}^2\sigma_x^2 + \eta_{y,l}^2\sigma_y^2\right).$$

*Proof.* Similar to the bound of $\Phi$ and $f$ in (2) and (3), we have the following results:

$$\mathbb{E}\Phi(\mathbf{x}_{t+1}) - \Phi(\mathbf{x}_t) \leq -\frac{1}{2}\eta_x K\|\nabla\Phi(\mathbf{x}_t)\|^2 - \frac{1}{4}\eta_x K\|\nabla_x f(\mathbf{z}_t)\|^2 + \frac{3}{2}\eta_x K\mathbb{E}\|\bar{\mathbf{e}}_{x,t}\|^2$$

$$+ \eta_x K \frac{L_f^2}{\mu^2} \left\| \nabla_y f(\mathbf{z}_t) \right\|^2 + \frac{1}{2} L \eta_x^2 K^2 \mathbb{E} \left\| \mathbf{u}_{x,t} - \mathbf{e}_{x,t} \right\|^2 .$$

$$f(\mathbf{z}_t) - \mathbb{E} f(\mathbf{z}_{t+1}) \le \frac{3}{2} \eta_x K \left\| \nabla_x f(\mathbf{z}_t) \right\|^2 + \frac{1}{2} \eta_x K \mathbb{E} \left\| \bar{\mathbf{e}}_{x,t} \right\|^2 + \frac{1}{2} \eta_y K \mathbb{E} \left\| \bar{\mathbf{e}}_{y,t} \right\|^2 - \frac{1}{2} \eta_y K \left\| \nabla_y f(\mathbf{z}_t) \right\|^2$$
$$+ \frac{1}{2} L_f \eta_x^2 K^2 \left\| \mathbf{u}_{x,t} - \mathbf{e}_{x,t} \right\|^2 + \frac{1}{2} L_f \eta_y^2 K^2 \left\| \mathbf{u}_{y,t} - \mathbf{e}_{y,t} \right\|^2 .$$

Define potential function $\mathcal{L}_t = \Phi(\mathbf{x}_t) - \frac{1}{10} f(\mathbf{z}_t)$,

$$\mathbb{E} \mathcal{L}_{t+1} - \mathcal{L}_t = \mathbb{E} \Phi(\mathbf{x}_{t+1}) - \Phi(\mathbf{x}_t) + \frac{1}{10} \left( f(\mathbf{z}_t) - \mathbb{E} f(\mathbf{z}_{t+1}) \right)$$

$$\le -\frac{1}{2} \eta_x K \|\nabla \Phi(\mathbf{x}_t)\|^2 - \frac{1}{10} \eta_x K \left\| \nabla_x f(\mathbf{z}_t) \right\|^2 - \eta_y K \left( \frac{1}{20} - \frac{\eta_x}{\eta_y} \frac{L_f^2}{\mu^2} \right) \left\| \nabla_y f(\mathbf{z}_t) \right\|^2$$

$$+ \frac{31}{20} \eta_x K \left\| \bar{\mathbf{e}}_{x,t} \right\|^2 + \frac{1}{20} \eta_y K \left\| \bar{\mathbf{e}}_{y,t} \right\|^2 + \frac{1}{2} \left( L + \frac{L_f}{10} \right) \eta_x^2 K^2 \mathbb{E} \left\| \Delta \mathbf{x}_t \right\|^2 + \frac{1}{20} L_f \eta_y^2 K^2 \mathbb{E} \left\| \Delta \mathbf{y}_t \right\|^2$$

$$\le -\frac{1}{2} \eta_x K \|\nabla \Phi(\mathbf{x}_t)\|^2 - \frac{1}{10} \eta_x K \left\| \nabla_x f(\mathbf{z}_t) \right\|^2 - \eta_y K \left( \frac{1}{20} - \frac{\eta_x}{\eta_y} \frac{L_f^2}{\mu^2} \right) \left\| \nabla_y f(\mathbf{z}_t) \right\|^2$$

$$+ K L_f^2 \left( \frac{31}{20} \eta_x + \frac{1}{20} \eta_y \right) \Psi_t + \frac{1}{2} \left( L + \frac{L_f}{10} \right) \mathbb{E} \left\| \mathbf{z}_{t+1} - \mathbf{z}_t \right\|^2$$

$$\left( \mathbb{E} \mathcal{L}_{t+1} + \alpha \Gamma_{t+1} \right) - \left( \mathcal{L}_t + \alpha \Gamma_t \right)$$

$$\le -\frac{1}{2} \eta_x K \|\nabla \Phi(\mathbf{x}_t)\|^2 - \frac{1}{10} \eta_x K \left\| \nabla_x f(\mathbf{z}_t) \right\|^2 - \eta_y K \left( \frac{1}{20} - \frac{\eta_x}{\eta_y} \frac{L_f^2}{\mu^2} \right) \left\| \nabla_y f(\mathbf{z}_t) \right\|^2$$

$$+ K L_f^2 \left( \frac{31}{20} \eta_x + \frac{1}{20} \eta_y \right) \Psi_t + \frac{1}{2} \left( L + \frac{L_f}{10} \right) \mathbb{E} \left\| \mathbf{z}_{t+1} - \mathbf{z}_t \right\|^2 + \alpha \Gamma_{t+1} - \alpha \Gamma_t$$

$$\le -\frac{1}{2} \eta_x K \|\nabla \Phi(\mathbf{x}_t)\|^2 - \frac{1}{10} \eta_x K \left\| \nabla_x f(\mathbf{z}_t) \right\|^2 - \eta_y K \left( \frac{1}{20} - \frac{\eta_x}{\eta_y} \frac{L_f^2}{\mu^2} \right) \left\| \nabla_y f(\mathbf{z}_t) \right\|^2$$

$$+ \underbrace{K L_f^2 \left( \frac{31}{20} \eta_x + \frac{1}{20} \eta_y \right)}_{a_1} \Psi_t + \underbrace{\left[ \frac{1}{2} \left( L + \frac{L_f}{10} \right) + \alpha \left( \frac{m}{M} + \frac{M}{m} - 1 \right) \right]}_{a_2} \mathbb{E} \left\| \mathbf{z}_{t+1} - \mathbf{z}_t \right\|^2 - \alpha \frac{m}{2M} \Gamma_t$$

$$\le -\frac{1}{2} \eta_x K \|\nabla \Phi(\mathbf{x}_t)\|^2 - \frac{1}{10} \eta_x K \left\| \nabla_x f(\mathbf{z}_t) \right\|^2 - \eta_y K \left( \frac{1}{20} - \frac{\eta_x}{\eta_y} \frac{L_f^2}{\mu^2} \right) \left\| \nabla_y f(\mathbf{z}_t) \right\|^2$$

$$+ \left[ a_1 + a_2 4 L_f^2 K^2 \left( \eta_x^2 + \eta_y^2 \right) \right] \Psi_t + \left[ 4 a_2 L_f^2 K^2 \left( \eta_x^2 + \eta_y^2 \right) - \alpha \frac{m}{2M} \right] \Gamma_t$$

$$+ a_2 \left[ 4 K^2 \left( \eta_x^2 \left\| \nabla_x f(\mathbf{z}_t) \right\|^2 + \eta_y^2 \left\| \nabla_y f(\mathbf{z}_t) \right\|^2 \right) + \frac{9K}{m} \left( \eta_x^2 \sigma_x^2 + \eta_y^2 \sigma_y^2 \right) \right]$$

$$\le -\frac{1}{2} \eta_x K \|\nabla \Phi(\mathbf{x}_t)\|^2 - \left[ \frac{1}{10} \eta_x K - 4 a_2 K^2 \eta_x^2 \right] \left\| \nabla_x f(\mathbf{z}_t) \right\|^2$$

$$- \left[ \eta_y K \left( \frac{1}{20} - \frac{\eta_x}{\eta_y} \frac{L_f^2}{\mu^2} \right) - 4 a_2 K^2 \eta_y^2 \right] \left\| \nabla_y f(\mathbf{z}_t) \right\|^2 + \left[ a_1 + a_2 4 L_f^2 K^2 \left( \eta_x^2 + \eta_y^2 \right) \right] \times$$

$$\left[ 10 K^2 \left( \eta_{x,l}^2 \sigma_x^2 + \eta_{y,l}^2 \sigma_y^2 \right) + 40 K^2 \left( \eta_{x,l}^2 \mathbb{E} \left\| \nabla_x f(\mathbf{z}_t) \right\|^2 + \eta_{y,l}^2 \mathbb{E} \left\| \nabla_y f(\mathbf{z}_t) \right\|^2 \right) \right]$$

$$- \left[ \alpha \frac{m}{2M} - 4 a_2 L_f^2 K^2 \left( \eta_x^2 + \eta_y^2 \right) - \left( a_1 + a_2 4 L_f^2 K^2 \left( \eta_x^2 + \eta_y^2 \right) \right) 160 K^2 \left( \eta_{x,l}^2 + \eta_{y,l}^2 \right) L_f^2 \right] \Gamma_t$$

$$+ a_2 \frac{9K}{m} \left( \eta_x^2 \sigma_x^2 + \eta_y^2 \sigma_y^2 \right),$$

where we can set $\alpha = \frac{M}{m}$ and requires the learning rates $\eta_x, \eta_y$ and $\eta_{x,l}, \eta_{y,l}$ satisfy

$$\left[\alpha\frac{m}{2M} - 4a_2 L_f^2 K^2 \left(\eta_x^2 + \eta_y^2\right) - \left(a_1 + a_2 4 L_f^2 K^2 \left(\eta_x^2 + \eta_y^2\right)\right) 160 K^2 \left(\eta_{x,l}^2 + \eta_{y,l}^2\right) L_f^2\right] \geq 0,$$

(11)

$$\left[\frac{1}{10}\eta_x K - 4a_2 K^2 \eta_x^2\right] - \left[a_1 + a_2 4 L_f^2 K^2 \left(\eta_x^2 + \eta_y^2\right)\right] 40 K^2 \eta_{x,l}^2 \geq 0,$$

(12)

$$\left[\eta_y K \left(\frac{1}{20} - \frac{\eta_x}{\eta_y}\frac{L_f^2}{\mu^2}\right) - 4a_2 K^2 \eta_y^2\right] - \left[a_1 + a_2 4 L_f^2 K^2 \left(\eta_x^2 + \eta_y^2\right)\right] 40 K^2 \eta_{y,l}^2 \geq 0.$$

(13)

$$(\mathbb{E}\mathcal{L}_{t+1} + \alpha\Gamma_{t+1}) - (\mathcal{L}_t + \alpha\Gamma_t)$$
$$\leq -\frac{1}{2}\eta_x K\|\nabla\Phi(\mathbf{x}_t)\|^2 + \left[a_1 + a_2 4 L_f^2 K^2 \left(\eta_x^2 + \eta_y^2\right)\right]\left[10 K^2 \left(\eta_{x,l}^2 \sigma_x^2 + \eta_{y,l}^2 \sigma_y^2\right)\right] + a_2 \frac{9K}{m}\left(\eta_x^2\sigma_x^2 + \eta_y^2\sigma_y^2\right)$$
$$\leq -\frac{1}{2}\eta_x K\|\nabla\Phi(\mathbf{x}_t)\|^2 + \left[\frac{1}{2}\left(L + \frac{L_f}{10}\right) + 2\right]\frac{9K}{m}\left(\eta_x^2\sigma_x^2 + \eta_y^2\sigma_y^2\right)$$
$$+ \left[KL_f^2\left(\frac{31}{20}\eta_x + \frac{1}{20}\eta_y\right) + \left[\frac{1}{2}\left(L + \frac{L_f}{10}\right) + 2\right]4L_f^2 K^2 \left(\eta_x^2 + \eta_y^2\right)\right]\left[10 K^2 \left(\eta_{x,l}^2\sigma_x^2 + \eta_{y,l}^2\sigma_y^2\right)\right]$$

Note that $\Gamma_0 = 0$.

Telescoping and rearranging, we have:

$$\frac{1}{T}\sum_{t=0}^{T-1}\mathbb{E}\|\nabla\Phi(\mathbf{x}_t)\|^2 \leq \frac{2(\mathcal{L}_0 - \mathcal{L}_*)}{\eta_x KT} + \left[\left(L + \frac{L_f}{10}\right) + 4\right]\frac{9}{m\eta_x}\left(\eta_x^2\sigma_x^2 + \eta_y^2\sigma_y^2\right)$$
$$+ \left[L_f^2\left(\frac{31}{20} + \frac{1}{20}\frac{\eta_y}{\eta_x}\right) + \left[\frac{1}{2}\left(L + \frac{L_f}{10}\right) + 2\right]4L_f^2 K\left(\eta_x + \frac{\eta_y^2}{\eta_x}\right)\right]\left[20 K^2\left(\eta_{x,l}^2\sigma_x^2 + \eta_{y,l}^2\sigma_y^2\right)\right].$$

$\square$

### B.3 Proof for SAGDA Option II

For SAGDA Option II, the update rule is:

$$\mathbf{x}_{t+1} = \mathbf{x}_t - \eta_{x,g}\eta_{x,l}\left[\frac{1}{m}\sum_{i\in S_t}\sum_{j\in[K]}\left(\nabla_x f_i(\mathbf{z}_{t,i}^j, \xi_{t,i}^j) - \nabla_x f_i(\mathbf{z}_t, \xi_{t,i}) + \frac{1}{m}\sum_{i\in S_t}\nabla_x f_i(\mathbf{z}_t, \xi_{t,i})\right)\right],$$

$$\mathbf{y}_{t+1} = \mathbf{y}_t + \eta_{y,g}\eta_{y,l}\left[\frac{1}{m}\sum_{i\in S_t}\sum_{j\in[K]}\left(\nabla_y f_i(\mathbf{z}_{t,i}^j, \xi_{t,i}^j) - \nabla_y f_i(\mathbf{z}_t, \xi_{t,i}) + \frac{1}{m}\sum_{i\in S_t}\nabla_y f_i(\mathbf{z}_t, \xi_{t,i})\right)\right],$$

$$\mathbf{e}_{x,t} = \frac{1}{mK}\sum_{i\in S_t}\sum_{j\in[K]}\left[\nabla_x f_i(\mathbf{z}_t) - \left(\nabla_x f_i(\mathbf{z}_{t,i}^j, \xi_{t,i}^j) - \nabla_x f_i(\mathbf{z}_t, \xi_{t,i}) + \frac{1}{m}\sum_{i\in S_t}\nabla_x f_i(\mathbf{z}_t, \xi_{t,i})\right)\right]$$
$$= \frac{1}{mK}\sum_{i\in S_t}\sum_{j\in[K]}\left[\nabla_x f_i(\mathbf{z}_t) - \nabla_x f_i(\mathbf{z}_{t,i}^j, \xi_{t,i}^j)\right],$$

$$\bar{\mathbf{e}}_{x,t} = \mathbb{E}[\mathbf{e}_{x,t}] = \frac{1}{mK}\sum_{i\in S_t}\sum_{j\in[K]}\left(\nabla_x f_i(\mathbf{z}_t) - \nabla_x f_i(\mathbf{z}_{t,i}^j)\right),$$

$$\mathbf{e}_{y,t} = \frac{1}{mK}\sum_{i\in S_t}\sum_{j\in[K]}\left[\nabla_y f_i(\mathbf{z}_t) - \left(\nabla_y f_i(\mathbf{z}_{t,i}^j, \xi_{t,i}^j) - \nabla_y f_i(\mathbf{z}_t, \xi_{t,i}) + \frac{1}{m}\sum_{i\in S_t}\nabla_y f_i(\mathbf{z}_t, \xi_{t,i})\right)\right]$$
$$= \frac{1}{mK}\sum_{i\in S_t}\sum_{j\in[K]}\left(\nabla_y f_i(\mathbf{z}_t) - \nabla_y f_i(\mathbf{z}_{t,i}^j, \xi_{t,i}^j)\right),$$

$$\bar{\mathbf{e}}_{y,t} = \mathbb{E}[\mathbf{e}_{y,t}] = \frac{1}{mK} \sum_{i \in S_t} \sum_{j \in [K]} \left( \nabla_y f_i(\mathbf{z}_t) - \nabla_y f_i(\mathbf{z}_{t,i}^j) \right).$$

**Lemma 9.**

$$\mathbb{E}\| \left( \mathbf{u}_{x,t} - \mathbf{e}_{x,t} \right) \|^2 \leq \frac{4}{MK} \sum_{i \in [M]} \sum_{j \in [K]} \left[ L_f^2 \mathbb{E} \left\| \mathbf{z}_{t,i}^j - \mathbf{z}_t \right\|^2 + \|\nabla_x f(\mathbf{z}_t)\|^2 \right] + \frac{9}{mK} \sigma_x^2,$$

$$\mathbb{E}\| \left( \mathbf{u}_{y,t} - \mathbf{e}_{y,t} \right) \|^2 \leq \frac{4}{MK} \sum_{i \in [M]} \sum_{j \in [K]} \left[ L_f^2 \mathbb{E} \left\| \mathbf{z}_{t,i}^j - \mathbf{z}_t \right\|^2 + \|\nabla_y f(\mathbf{z}_t)\|^2 \right] + \frac{9}{mK} \sigma_y^2.$$

*Proof.*

$$\mathbb{E}\| \left( \mathbf{u}_{x,t} - \mathbf{e}_{x,t} \right) \|^2 \leq \mathbb{E} \left\| \frac{1}{mK} \sum_{i \in S_t} \sum_{j \in [K]} \left[ \nabla_x f_i(\mathbf{z}_{t,i}^j) - \mathbb{E}[\mathbf{v}_{x,t}^i] + \mathbb{E}[\bar{\mathbf{v}}_{x,t}] \right] \right\|^2 + \frac{9}{mK} \sigma_x^2$$

$$\leq \frac{4}{MK} \sum_{i \in [M]} \sum_{j \in [K]} \left[ \mathbb{E} \left\| \nabla_x f_i(\mathbf{z}_{t,i}^j) - \nabla_x f_i(\mathbf{z}_t) \right\|^2 + \mathbb{E} \left\| \mathbb{E}[\mathbf{v}_{x,t}^i] - \nabla_x f_i(\mathbf{z}_t) \right\|^2 \right.$$

$$\left. + \mathbb{E} \left\| \mathbb{E}[\bar{\mathbf{v}}_{x,t}] - \nabla_x f(\mathbf{z}_t) \right\|^2 + \|\nabla_x f(\mathbf{z}_t)\|^2 \right] + \frac{9}{mK} \sigma_x^2$$

$$\leq \frac{4}{MK} \sum_{i \in [M]} \sum_{j \in [K]} \left[ L_f^2 \mathbb{E} \left\| \mathbf{z}_{t,i}^j - \mathbf{z}_t \right\|^2 + \|\nabla_x f(\mathbf{z}_t)\|^2 \right] + \frac{9}{mK} \sigma_x^2,$$

where the last inequality is due to $\mathbb{E}[\mathbf{v}_{x,t}^i] = \nabla_x f_i(\mathbf{z}_t)$ and $\mathbb{E}[\bar{\mathbf{v}}_{x,t}] = \nabla_x f(\mathbf{z}_t)$, and the second inequality is due to Lemma 4 in [28]).

The bound of $\| \left( \mathbf{u}_{y,t} - \mathbf{e}_{y,t} \right) \|^2$ follows from the similar proof. $\qquad\square$

**Lemma 10** (Bounded Error for SAGDA Option II)**.**

$$\mathbb{E}\|\bar{\mathbf{e}}_{x,t}\|^2 \quad \leq \frac{L_f^2}{MK} \sum_{i \in [M], j \in [K]} \mathbb{E} \left\| \left( \mathbf{z}_t - \mathbf{z}_{t,i}^j \right) \right\|^2,$$

$$\mathbb{E}\|\bar{\mathbf{e}}_{y,t}\|^2 \quad \leq \frac{L_f^2}{MK} \sum_{i \in [M], j \in [K]} \mathbb{E} \left\| \left( \mathbf{z}_t - \mathbf{z}_{t,i}^j \right) \right\|^2.$$

*Proof.*

$$\mathbb{E}\|\bar{\mathbf{e}}_{x,t}\|^2 = \mathbb{E} \left\| \frac{1}{mK} \sum_{i \in S_t} \sum_{j \in [K]} \left( \nabla_x f_i(\mathbf{z}_t) - \nabla_x f_i(\mathbf{z}_{t,i}^j) \right) \right\|^2$$

$$\leq \frac{1}{mK} \mathbb{E} \left[ \sum_{i \in S_t} \sum_{j \in [K]} \left\| \left( \nabla_x f_i(\mathbf{z}_t) - \nabla_x f_i(\mathbf{z}_{t,i}^j) \right) \right\|^2 \right]$$

$$\leq \frac{L_f^2}{MK} \sum_{i \in [M], j \in [K]} \mathbb{E} \left\| \left( \mathbf{z}_t - \mathbf{z}_{t,i}^j \right) \right\|^2.$$

$\mathbb{E}\|\bar{\mathbf{e}}_{y,t}\|^2$ has the same bounds. $\qquad\square$

**Lemma 11** (Local Step Distance for SAGDA Option II)**.** $\forall i \in [M], j \in [K]$, *we can bound the local step distance as follows:*

$$\mathbb{E} \left\| \left( \mathbf{z}_t - \mathbf{z}_{t,i}^j \right) \right\|^2$$

$$\leq 5K \left( 16K + 1 \right) \eta_{x,l}^2 \sigma_x^2 + 5K \left( 16K + 1 \right) \eta_{y,l}^2 \sigma_y^2 + 40K^2 \left( \eta_{x,l}^2 \mathbb{E} \|\nabla_x f(\mathbf{z}_t)\|^2 + \eta_{y,l}^2 \mathbb{E} \|\nabla_y f(\mathbf{z}_t)\|^2 \right).$$

*Proof.*

$$\mathbb{E}\left\|\left(\mathbf{z}_t - \mathbf{z}_{t,i}^{j+1}\right)\right\|^2 = \mathbb{E}\left[\left\|\mathbf{x}_{t,i}^j - \mathbf{x}_t - \eta_{x,l}\left(\nabla_x f_i(\mathbf{z}_{t,i}^j, \xi_{t,i}^j) - \mathbf{v}_{x,t}^i + \bar{\mathbf{v}}_{x,t}\right)\right\|^2\right]$$

$$+ \mathbb{E}\left[\left\|\mathbf{y}_{t,i}^j - \mathbf{y}_t + \eta_{y,l}\left(\nabla_y f_i(\mathbf{z}_{t,i}^j, \xi_{t,i}^j) - \mathbf{v}_{x,t}^i + \bar{\mathbf{v}}_{y,t}\right)\right\|^2\right]$$

$$= \mathbb{E}\left[\left\|\mathbf{x}_{t,i}^j - \mathbf{x}_t - \eta_{x,l}\left(\nabla_x f_i(\mathbf{z}_{t,i}^j) - \mathbf{v}_{x,t}^i + \bar{\mathbf{v}}_{x,t}\right)\right\|^2\right] + \eta_{x,l}^2 \sigma_x^2$$

$$+ \mathbb{E}\left[\left\|\mathbf{y}_{t,i}^j - \mathbf{y}_t + \eta_{y,l}\left(\nabla_y f_i(\mathbf{z}_{t,i}^j) - \mathbf{v}_{x,t}^i + \bar{\mathbf{v}}_{y,t}\right)\right\|^2\right] + \eta_{y,l}^2 \sigma_y^2$$

$$= \left(1 + \frac{1}{2K-1}\right)\mathbb{E}\left\|\mathbf{x}_{t,i}^j - \mathbf{x}_t\right\|^2 + 2K\mathbb{E}\left\|\eta_{x,l}\left(\nabla_x f_i(\mathbf{z}_{t,i}^j) - \mathbf{v}_{x,t}^i + \bar{\mathbf{v}}_{x,t}\right)\right\|^2 + \eta_{x,l}^2 \sigma_x^2$$

$$+ \left(1 + \frac{1}{2K-1}\right)\mathbb{E}\left\|\mathbf{y}_{t,i}^j - \mathbf{y}_t\right\|^2 + 2K\mathbb{E}\left\|\eta_{y,l}\left(\nabla_y f_i(\mathbf{z}_{t,i}^j) - \mathbf{v}_{x,t}^i + \bar{\mathbf{v}}_{y,t}\right)\right\|^2 + \eta_{y,l}^2 \sigma_y^2$$

$$= \left(1 + \frac{1}{2K-1}\right)\mathbb{E}\left\|\mathbf{z}_{t,i}^j - \mathbf{z}_t\right\|^2 + 2K\mathbb{E}\left\|\eta_{x,l}\left(\nabla_x f_i(\mathbf{z}_{t,i}^j) - \mathbf{v}_{x,t}^i + \bar{\mathbf{v}}_{x,t}\right)\right\|^2 + \eta_{x,l}^2 \sigma_x^2$$

$$+ 2K\mathbb{E}\left\|\eta_{y,l}\left(\nabla_y f_i(\mathbf{z}_{t,i}^j) - \mathbf{v}_{x,t}^i + \bar{\mathbf{v}}_{y,t}\right)\right\|^2 + \eta_{y,l}^2 \sigma_y^2$$

$$\leq \left(1 + \frac{1}{2K-1}\right)\mathbb{E}\left\|\mathbf{z}_{t,i}^j - \mathbf{z}_t\right\|^2 + 2K\eta_{x,l}^2\left[4L_f^2\mathbb{E}\left\|\mathbf{z}_{t,i}^j - \mathbf{z}_t\right\|^2 + 8\sigma_x^2 + 4\mathbb{E}\left\|\nabla_x f(\mathbf{z}_t)\right\|^2\right]$$

$$+ 2K\eta_{y,l}^2\left[4L_f^2\mathbb{E}\left\|\mathbf{z}_{t,i}^j - \mathbf{z}_t\right\|^2 + 8\sigma_y^2 + 4\mathbb{E}\left\|\nabla_y f(\mathbf{z}_t)\right\|^2\right] + \eta_{x,l}^2 \sigma_x^2 + \eta_{y,l}^2 \sigma_y^2$$

$$\leq \left(1 + \frac{1}{2K-1} + 8K\max\{L_f^2\eta_{x,l}^2, L_f^2\eta_{y,l}^2\}\right)\mathbb{E}\left\|\mathbf{z}_{t,i}^j - \mathbf{z}_t\right\|^2 + (16K+1)\eta_{x,l}^2 \sigma_x^2$$

$$+ (16K+1)\eta_{y,l}^2 \sigma_y^2 + 8K\left(\eta_{x,l}^2\mathbb{E}\left\|\nabla_x f(\mathbf{z}_t)\right\|^2 + \eta_{y,l}^2\mathbb{E}\left\|\nabla_y f(\mathbf{z}_t)\right\|^2\right)$$

$$\leq \left(1 + \frac{1}{K-1}\right)\mathbb{E}\left\|\mathbf{z}_{t,i}^j - \mathbf{z}_t\right\|^2 + (16K+1)\eta_{x,l}^2 \sigma_x^2$$

$$+ (16K+1)\eta_{y,l}^2 \sigma_y^2 + 8K\left(\eta_{x,l}^2\mathbb{E}\left\|\nabla_x f(\mathbf{z}_t)\right\|^2 + \eta_{y,l}^2\mathbb{E}\left\|\nabla_y f(\mathbf{z}_t)\right\|^2\right)$$

$$\leq \sum_{\tau=0}^{j-1}\left(1 + \frac{1}{K-1}\right)^\tau\left[(16K+1)\eta_{x,l}^2 \sigma_x^2 + (16K+1)\eta_{y,l}^2 \sigma_y^2\right.$$

$$\left. + 8K\left(\eta_{x,l}^2\mathbb{E}\left\|\nabla_x f(\mathbf{z}_t)\right\|^2 + \eta_{y,l}^2\mathbb{E}\left\|\nabla_y f(\mathbf{z}_t)\right\|^2\right)\right]$$

$$\leq 5K(16K+1)\eta_{x,l}^2 \sigma_x^2 + 5K(16K+1)\eta_{y,l}^2 \sigma_y^2 + 40K^2\left(\eta_{x,l}^2\mathbb{E}\left\|\nabla_x f(\mathbf{z}_t)\right\|^2 + \eta_{y,l}^2\mathbb{E}\left\|\nabla_y f(\mathbf{z}_t)\right\|^2\right),$$

$\bar{\mathbf{v}}_{x,t} = \frac{1}{m}\sum_{i \in S_t}\nabla_x f_i(\mathbf{z}_t, \xi_{t,i})$ and $\mathbf{v}_{x,t}^i = \nabla_x f_i(\mathbf{z}_t, \xi_{t,i})$; $\bar{\mathbf{v}}_{y,t} = \frac{1}{m}\sum_{i \in S_t}\nabla_y f_i(\mathbf{z}_t, \xi_{t,i})$ and $\bar{\mathbf{v}}_{y,t}^i = \nabla_y f_i(\mathbf{z}_t, \xi_{t,i})$; where the first inequality is due to bounded variance of stochastic gradient, the second and third inequalities follow from the fact $\|\mathbf{a} + \mathbf{b}\|^2 \leq \left(1 + \frac{1}{\epsilon}\right)\|\mathbf{a}\|^2 + (1 + \epsilon)\|\mathbf{b}\|^2$, the forth inequality is due to smoothness of $f$ in $x$ and $y$, fifth inequality holds if

$$4K\max\{L_f^2\eta_{x,l}^2, L_f^2\eta_{y,l}^2\} \leq \frac{1}{2(K-1)(2K-1)}, \tag{14}$$

and the last inequality follows from the $\sum_{\tau=0}^{j-1}\left(1 + \frac{1}{K-1}\right)^\tau \leq (K-1)\left[\left(1 + \frac{1}{K-1}\right)^K - 1\right] \leq 5K$.

$$\mathbb{E}\left\|\left(\nabla_x f_i(\mathbf{z}_{t,i}^j) - \mathbf{v}_{x,t}^i + \bar{\mathbf{v}}_{x,t}\right)\right\|^2$$

$$= \mathbb{E}\left\|\left(\nabla_x f_i(\mathbf{z}_{t,i}^j) - \nabla_x f_i(\mathbf{z}_t)\right) + \left(\nabla_x f_i(\mathbf{z}_t) - \mathbf{v}_{x,t}^i\right) + (\bar{\mathbf{v}}_{x,t} - \nabla_x f(\mathbf{z}_t)) + \nabla_x f(\mathbf{z}_t)\right\|^2$$

$$\leq 4\mathbb{E}\left\|\nabla_x f_i(\mathbf{z}_{t,i}^j) - \nabla_x f_i(\mathbf{z}_t)\right\|^2 + 4\mathbb{E}\left\|\nabla_x f_i(\mathbf{z}_t) - \mathbf{v}_{x,t}^i\right\|^2 + 4\mathbb{E}\left\|\bar{\mathbf{v}}_{x,t} - \nabla_x f(\mathbf{z}_t)\right\|^2 + 4\mathbb{E}\left\|\nabla_x f(\mathbf{z}_t)\right\|^2$$

$$\leq 4L_f^2 \mathbb{E}\left\|\mathbf{z}_{t,i}^j - \mathbf{z}_t\right\|^2 + 8\sigma_x^2 + 4\mathbb{E}\left\|\nabla_x f(\mathbf{z}_t)\right\|^2$$

$\square$

*Proof.* Similar to the bound of $\Phi$ and $f$ in (2) and (3), we have the following results:

$$\mathbb{E}\Phi(\mathbf{x}_{t+1}) - \Phi(\mathbf{x}_t) \leq -\frac{1}{2}\eta_x K \|\nabla\Phi(\mathbf{x}_t)\|^2 - \frac{1}{4}\eta_x K \|\nabla_x f(\mathbf{z}_t)\|^2 + \frac{3}{2}\eta_x K \mathbb{E}\left\|\bar{\mathbf{e}}_{x,t}\right\|^2$$
$$+ \eta_x K \frac{L_f^2}{\mu^2}\left\|\nabla_y f(\mathbf{z}_t)\right\|^2 + \frac{1}{2}L\eta_x^2 K^2 \mathbb{E}\left\|\mathbf{u}_{x,t} - \mathbf{e}_{x,t}\right\|^2.$$

$$f(\mathbf{z}_t) - \mathbb{E}f(\mathbf{z}_{t+1}) \leq \frac{3}{2}\eta_x K \left\|\nabla_x f(\mathbf{z}_t)\right\|^2 + \frac{1}{2}\eta_x K \mathbb{E}\left\|\bar{\mathbf{e}}_{x,t}\right\|^2 + \frac{1}{2}\eta_y K \mathbb{E}\left\|\bar{\mathbf{e}}_{y,t}\right\|^2 - \frac{1}{2}\eta_y K \|\nabla_y f(\mathbf{z}_t)\|^2$$
$$+ \frac{1}{2}L_f \eta_x^2 K^2 \left\|\mathbf{u}_{x,t} - \mathbf{e}_{x,t}\right\|^2 + \frac{1}{2}L_f \eta_y^2 K^2 \left\|\mathbf{u}_{y,t} - \mathbf{e}_{y,t}\right\|^2.$$

Define potential function $\mathcal{L}_t = \Phi(\mathbf{x}_t) - \frac{1}{10}f(\mathbf{z}_t)$,

$$\mathbb{E}\mathcal{L}_{t+1} - \mathcal{L}_t = \mathbb{E}\Phi(\mathbf{x}_{t+1}) - \Phi(\mathbf{x}_t) + \frac{1}{10}\left(f(\mathbf{z}_t) - \mathbb{E}f(\mathbf{z}_{t+1})\right)$$

$$\leq -\frac{1}{2}\eta_x K\|\nabla\Phi(\mathbf{x}_t)\|^2 - \frac{1}{10}\eta_x K\left\|\nabla_x f(\mathbf{z}_t)\right\|^2 - \eta_y K\left(\frac{1}{20} - \frac{\eta_x}{\eta_y}\frac{L_f^2}{\mu^2}\right)\|\nabla_y f(\mathbf{z}_t)\|^2$$
$$+ \frac{31}{20}\eta_x K \|\bar{\mathbf{e}}_{x,t}\|^2 + \frac{1}{20}\eta_y K \|\bar{\mathbf{e}}_{y,t}\|^2$$
$$+ \frac{1}{2}\left(L + \frac{L_f}{10}\right)\eta_x^2 K^2 \mathbb{E}\left\|\mathbf{u}_{x,t} - \mathbf{e}_{x,t}\right\|^2 + \frac{1}{20}L_f \eta_y^2 K^2 \mathbb{E}\left\|\mathbf{u}_{y,t} - \mathbf{e}_{y,t}\right\|^2$$

$$\leq -\frac{1}{2}\eta_x K\|\nabla\Phi(\mathbf{x}_t)\|^2 - \frac{1}{10}\eta_x K\left\|\nabla_x f(\mathbf{z}_t)\right\|^2 - \eta_y K\left(\frac{1}{20} - \frac{\eta_x}{\eta_y}\frac{L_f^2}{\mu^2}\right)\|\nabla_y f(\mathbf{z}_t)\|^2$$
$$+ \left(\frac{31}{20}\eta_x K + \frac{1}{20}\eta_y K\right)\left[\frac{L_f^2}{MK}\sum_{i\in[M],j\in[K]}\mathbb{E}\left\|\left(\mathbf{z}_t - \mathbf{z}_{t,i}^j\right)\right\|^2\right]$$
$$+ \frac{1}{2}\left(L + \frac{L_f}{10}\right)\eta_x^2 K^2\left[\frac{4}{MK}\sum_{i\in[M]}\sum_{j\in[K]}\left[L_f^2\mathbb{E}\left\|\mathbf{z}_{t,i}^j - \mathbf{z}_t\right\|^2 + \|\nabla_x f(\mathbf{z}_t)\|^2\right] + \frac{9}{mK}\sigma_x^2\right]$$
$$+ \frac{1}{20}L_f \eta_y^2 K^2\left[\frac{4}{MK}\sum_{i\in[M]}\sum_{j\in[K]}\left[L_f^2\mathbb{E}\left\|\mathbf{z}_{t,i}^j - \mathbf{z}_t\right\|^2 + \|\nabla_y f(\mathbf{z}_t)\|^2\right] + \frac{9}{mK}\sigma_y^2\right]$$

$$\leq -\frac{1}{2}\eta_x K\|\nabla\Phi(\mathbf{x}_t)\|^2 - \frac{1}{10}\eta_x K\left\|\nabla_x f(\mathbf{z}_t)\right\|^2 - \eta_y K\left(\frac{1}{20} - \frac{\eta_x}{\eta_y}\frac{L_f^2}{\mu^2}\right)\|\nabla_y f(\mathbf{z}_t)\|^2$$
$$+ \underbrace{L_f^2\left[\frac{31}{20}\eta_x K + \frac{1}{20}\eta_y K + 2\left(L + \frac{L_f}{10}\right)\eta_x^2 K^2 + \frac{1}{5}L_f\eta_y^2 K^2\right]}_{a_1}\left[\frac{1}{MK}\sum_{i\in[M],j\in[K]}\mathbb{E}\left\|\left(\mathbf{z}_t - \mathbf{z}_{t,i}^j\right)\right\|^2\right]$$
$$+ \underbrace{2\left(L + \frac{L_f}{10}\right)\eta_x^2 K^2}_{a_2}\|\nabla_x f(\mathbf{z}_t)\|^2 + \frac{1}{2}\left(L + \frac{L_f}{10}\right)\eta_x^2 K^2\frac{9}{mK}\sigma_x^2$$
$$+ \underbrace{\frac{1}{5}L_f\eta_y^2 K^2}_{a_3}\|\nabla_y f(\mathbf{z}_t)\|^2 + \frac{1}{20}L_f\eta_y^2 K^2\frac{9}{mK}\sigma_y^2$$

$$\leq -\frac{1}{2}\eta_x K \|\nabla\Phi(\mathbf{x}_t)\|^2 - \frac{1}{10}\eta_x K \|\nabla_x f(\mathbf{z}_t)\|^2 - \eta_y K \left(\frac{1}{20} - \frac{\eta_x}{\eta_y}\frac{L_f^2}{\mu^2}\right)\|\nabla_y f(\mathbf{z}_t)\|^2$$

$$+ \left[5K(16K+1)\eta_{x,l}^2 a_1 + \frac{1}{2}\left(L + \frac{L_f}{10}\right)\eta_x^2 \frac{9K}{m}\right]\sigma_x^2 + \left[5K(16K+1)\eta_{y,l}^2 a_1 + \frac{1}{20}L_f\eta_y^2 \frac{9K}{m}\right]\sigma_y^2$$

$$+ \left(a_2 + 40K^2\eta_{x,l}^2 a_1\right)\|\nabla_x f(\mathbf{z}_t)\|^2 + \left(a_3 + 40K^2\eta_{y,l}^2 a_1\right)\|\nabla_y f(\mathbf{z}_t)\|^2$$

$$\leq -\frac{1}{2}\eta_x K\|\nabla\Phi(\mathbf{x}_t)\|^2 + \left[5K(16K+1)\eta_{x,l}^2 a_1 + \frac{1}{2}\left(L + \frac{L_f}{10}\right)\eta_x^2 \frac{9K}{m}\right]\sigma_x^2$$

$$+ \left[5K(16K+1)\eta_{y,l}^2 a_1 + \frac{1}{20}L_f\eta_y^2 \frac{9K}{m}\right]\sigma_y^2$$

where the last inequality follows from the conditions:

$$\frac{1}{10}\eta_x K - \left(a_2 + 40K^2\eta_{x,l}^2 a_1\right) \geq 0, \tag{15}$$

$$\eta_y K \left(\frac{1}{20} - \frac{\eta_x}{\eta_y}\frac{L_f^2}{\mu^2}\right) - \left(a_3 + 40K^2\eta_{y,l}^2 a_1\right) \geq 0. \tag{16}$$

Telescoping and rearranging, we have:

$$\frac{1}{T}\sum_{t=0}^{T-1}\mathbb{E}\|\nabla\Phi(\mathbf{x}_t)\|^2 \leq \frac{2(\mathcal{L}_0 - \mathcal{L}_*)}{\eta_x KT} + \left[10(16K+1)\eta_{x,l}^2\frac{a_1}{\eta_x} + \left(L + \frac{L_f}{10}\right)\frac{9\eta_x}{m}\right]\sigma_x^2$$

$$+ \left[10(16K+1)\eta_{y,l}^2\frac{a_1}{\eta_x} + \frac{9}{10}L_f\frac{\eta_y^2}{m\eta_x}\right]\sigma_y^2$$

$$\leq \frac{2(\mathcal{L}_0 - \mathcal{L}_*)}{\eta_x KT} + \left[\left(L + \frac{L_f}{10}\right)\frac{9\eta_x}{m}\sigma_x^2 + + \frac{9}{10}L_f\frac{\eta_y^2}{m\eta_x}\sigma_y^2\right]$$

$$+ L_f^2\left[\frac{31}{20}K + \frac{1}{20}\frac{\eta_y}{\eta_x}K + 2\left(L + \frac{L_f}{10}\right)\eta_x K^2 + \frac{1}{5}L_f\frac{\eta_y^2}{\eta_x}K^2\right][10(16K+1)]\left(\eta_{x,l}^2\sigma_x^2 + \eta_{y,l}^2\sigma_y^2\right).$$

$$\square$$