# OpenReview forum: "SAGDA: Achieving $\mathcal{O}(\epsilon^{-2})$ Communication Complexity in Federated Min-Max Learning"
_NeurIPS.cc/2022/Conference — NeurIPS 2022 Accept_

### Official Review · Reviewer_YJdZ · 2022-07-07

**Rating:** 6
**Confidence:** 2
**Soundness:** 3 good
**Presentation:** 2 fair
**Contribution:** 4 excellent

**Summary:**

The paper provides a new algorithm, SAGDA, for federated learning in min-max learning, and proves a theoretical communication complexity guarantee that significantly improves upon previous results (without harming sample complexity). The main idea of SAGDA is to leverage new control variates, and to update a sample of them, based on the variations of the clients’ gradients. In particular, the control variates are used by the clients to update their local models, thereby avoiding model drift. Interestingly, this also allows the authors to remove the bounded gradient dissimilarity assumption, which was needed in prior works. Empirical results also comfort the effectiveness of SAGDA.

The paper also shows the same communication complexity for the standard FSGDA algorithm, by proving that FSGDA is a special case of SAGDA where the control covariates are canceled.


**Questions:**

It is not clear to me if the techniques of the paper apply to minimization learning in general (and not just min-max learning). Could SAGDA be easily generalized into SAGD? (in particular, the use of control variates)

If not, it would be interesting to discuss why not.

**Limitations:**

The paper is fully dedicated to machine learning performance, under the assumption that it is desirable to generalize collected data. This clearly raises major societal concerns, if the collected data is undesirably biased, or if it is harmed by poisoning data attacks. Similarly, since it relies on averaging clients’ inputs, SAGDA is arbitrarily manipulable by malicious clients. On the Internet, such malicious clients are ubiquitous.

I believe that the authors have a responsibility to highlight this, and to warn against the use of such algorithms in environments where the quality of the data is not sufficiently guaranteed, or when the clients are not all extremely trustworthy. A discussion on the need for developing Byzantine-resilient variants of SAGDA would be ethically productive.


**Strengths And Weaknesses:**

Given the increasing cost of communications in federated learning, the paper provides an original and significant contribution to the design and understanding of efficient federated learning algorithms.

I have however found the paper hard to read, especially as the authors aim to present two variants of SAGDA at once. I would encourage the authors to first focus on only one flavor of SAGDA and to make it as clear as possible for the reader. The other variant may then be briefly discussed in the main paper, and described more thoroughly in the appendix. Moreover, I would have found it useful to also include Equations (2) and (3) directly in the algorithm.

I also feel that too much space is dedicated to the case of FSGDA, which seems less important to me than SAGDA. I would have found it more valuable to dedicate more space to explaining the intuition of SAGDA.

---

> ### Author Response · Authors · 2022-08-02
> **Response to Reviewer YJdZ [1/2]**
>
> Thank you very much for your review and constructive comments, which helped us significantly improve the quality of this paper. In this revision, we have carefully revised our paper based on your comments, questions, and suggestions. The detailed point-by-point responses are as follows:
>
> ### **Weaknesses**
>
> > 1. I have however found the paper hard to read, especially as the authors aim to present two variants of SAGDA at once. I would encourage the authors to first focus on only one flavor of SAGDA and to make it as clear as possible for the reader. The other variant may then be briefly discussed in the main paper, and described more thoroughly in the appendix. Moreover, I would have found it useful to also include Equations (2) and (3) directly in the algorithm.
>
> **Our response:** Thanks for your suggestions. The reason that we present two variants of SAGDA in the same algorithmic framework is that both variants have their own benefits and neither dominates the other. As discussed in the paper (cf. Lines 152--161), Option I requires the clients to be stateful and has a higher implementation complexity, while Option II may incur some extra communication overhead due to more communication sessions are needed. In our opinion, both variants have their own merits and deserve equal attention. But we do agree with the reviewer that the clarity of the presentation could be more reader-friendly, and we will improve this in the revision.
>
> > 2. I also feel that too much space is dedicated to the case of FSGDA, which seems less important to me than SAGDA. I would have found it more valuable to dedicate more space to explaining the intuition of SAGDA.
>
> **Our response:** Thanks for your comments and suggestions. We remark that the discussion of FSGDA is also important because our results for FSGDA arrived through the specialization of SAGDA provides new insights and advances the current understanding of FSGDA -- a popular and important basic method in the literature for federated min-max learning. Due to this reason, the discussion of FSGDA through the lens of SAGDA deserves its place in the main paper. But we do agree with the reviewer that we could plan the space more wisely, and we will improve this in our revision.
>
> Also, we would like to take this opportunity to futher clarify the ideas behind SAGDA, which stems from two key aspects: 1) the use of control variates in the local update to effectively mitigate the data heterogeneity, and thus removing any assumption on the non-i.i.d. datasets; 2) we utilize two-sided learning rates for the updates of both parameters $x$ and $y$. Specifically, we use a small local learning rate can properly control the model discrepancy among clients and a relatively large global learning rate for aggregation. We will add the above discussions on the basic idea of SAGDA in the revision.
>
> ### **Question**
>
> > 1. It is not clear to me if the techniques of the paper apply to minimization learning in general (and not just min-max learning). Could SAGDA be easily generalized into SAGD? (in particular, the use of control variates). If not, it would be interesting to discuss why not.
>
> **Our response:** Thanks for your comments and questions. The answer is ``yes,'' i.e., SAGDA could be generalized into SAGD for minimization problems. The basic idea is again to utilize the approximated stochastic gradient as the control variates to tackle data heterogeneity. In doing so, we only need to keep the update for $x$ and $v_x$, while removing the part for $y$ and $v_y$. Similar convergence proof techniques are still applicable to show similar convergence rate results, which further implies similar improved sample and communication complexities. In this sense, the results and techniques in this paper could be of independent interests. Thanks for pointing this out.

---

> > ### Author Response · Authors · 2022-08-02
> > **Response to Reviewer YJdZ [2/2]**
> >
> > ### **Limitations**
> >
> > > 1. The paper is fully dedicated to machine learning performance, under the assumption that it is desirable to generalize collected data. This clearly raises major societal concerns, if the collected data is undesirably biased, or if it is harmed by poisoning data attacks. Similarly, since it relies on averaging clients’ inputs, SAGDA is arbitrarily manipulable by malicious clients. On the Internet, such malicious clients are ubiquitous.
> > I believe that the authors have a responsibility to highlight this, and to warn against the use of such algorithms in environments where the quality of the data is not sufficiently guaranteed, or when the clients are not all extremely trustworthy. A discussion on the need for developing Byzantine-resilient variants of SAGDA would be ethically productive.}
> >
> > **Our response:** Thanks for your suggestions. We agree with you that discussing the societal concerns and potentially negative impacts of the SAGDA algorithm is important. In our revision, we will add an extra section to discuss and highlight such societal concerns and warnings, including trustworthiness of the data collection, the need for Byzantine-resilient variants of SAGDA, etc.

---

### Official Review · Reviewer_oLiX · 2022-07-13

**Rating:** 6
**Confidence:** 3
**Soundness:** 3 good
**Presentation:** 3 good
**Contribution:** 3 good

**Summary:**

-----After rebuttal

I thank the author for answering my two questions. Given my assessment of the novelty and significance of the paper, I am keeping the score as is.

-----Original review

This paper studies the federated min-max problems which has applications to distributed robust optimization for instance. Two federated versions of the celebrated SGDA algorithm are explored by leveraging standard technique. SAGDA which uses global variance reduction by employing control variates and FSGDA which only employs client sampling and local updates (the latter is an existing method). The convergence of these methods are analyzed for nonconvex-PL min-max problems and it is shown they achieve similar iteration and communication complexity as those of FedAvg for federated minimization problem.

**Questions:**

1. It seems the CD-MAGE+ algorithm achieves a superior sample and communication complexity compared to the result of the present paper. Can you elaborate on the comparison of the two? (answered in rebuttal)

2. The title is misleading as $O(\epsilon^{-2})$ is already achieved by the prior work.  (answered in rebuttal)

**Limitations:**

yes

**Strengths And Weaknesses:**

Strength:
1. Improving sample and communication complexity over [14]

Weakness
1. The results while theoretically appleaning does not seems surprising. The proposed algorithm and the proofs seem to be an application of the known techniques in federated and stochastic optimization.

---

> ### Author Response · Authors · 2022-08-02
> **Response to Reviewer oLiX**
>
> Thank you very much for your review and constructive comments, which helped us significantly improve the quality of this paper. In this revision, we have carefully revised our paper based on your comments, questions, and suggestions. The detailed point-by-point responses are as follows:
>
> ### **Weakness**
>
> > 1. The results while theoretically appleaning does not seems surprising. The proposed algorithm and the proofs seem to be an application of the known techniques in federated and stochastic optimization.
>
> **Our response:**
> Thanks for your comments.
> Although there exist a few papers studying federated learning and stochastic min-max optimization, federated min-max learning is still an important question and calls for investigations.
> The key novelty of our work stems from solving federated min-max learning with two main challenges: 1) non-i.i.d. datasets and 2) partial client participation.
> In the previous literature, most of the federated min-max learning algorithms rely on the BGD assumption.
> In our work, we proposed a new algorithmic framework named SAGDA to address the above challenges.
> Our algorithm leverages control-variate and two-sided learning rates techniques to handle non-i.i.d data and stabilize FL training, respectively.
> These techniques were originally developed for federated minimization learning, however their performances on min-max problem remain unclear.
> In our work, we filled the gap and studied the theoretical convergence performance of SAGDA with the above two techniques on federated min-max problems.
> We showed that SAGDA achieves a linear speedup in terms of both the number of clients and local update steps.
> Also SAGDA allows the local update round $K$ to be on the order of $\mathcal{O}(T/m)$, which further implies an $O(\epsilon^{-2})$ communication complexity.
> To our knowledge, the communication complexity of our SAGDA algorithm is much better than the-state-of-the-art results in the literature.
>
> ### **Questions**
> > 1. It seems the CD-MAGE+ algorithm achieves a superior sample and communication complexity compared to the result of the present paper. Can you elaborate on the comparison of the two?
>
> **Our response:** Thanks for your comments. For communication complexity, CD-MAGE+ requires $\mathcal{O}(\epsilon^{-3})$ rounds to achieve an $\epsilon$-stationary point (i.e., $\| \nabla \Phi \| \leq \epsilon$ ), and its convergence rate does not suggest any convergence speedup in terms of $K$.
> Our SAGDA algorithm achieves $\mathcal{O}(\epsilon^{-2})$, which enjoys a better communication complexity. For sample complexity, our SAGDA algorithm achieves $\mathcal{O}(m^{-1} \epsilon^{-4})$, which matches the state-of-the-art rates in federated min-max problems (see Table 1 in [15]). CD-MAGE+ achieves $\mathcal{O}(m^{-1/2} \epsilon^{-3})$ based on their convergence rate (Thm 2 and remark 3 in their paper). They have a better dependence on $\epsilon$ but a worse dependence on $m$, hence lacking the benefit of linear speedup with respect to $m$. We believe that the reason is due to their use of STORM-type updates ([R1]) for the local gradient estimator.
>
> > 2. The title is misleading as $\mathcal{O}(\epsilon^{-2})$ is already achieved by the prior work.
>
> **Our response:** Thanks for your comments. So far, we are not aware of any previous work that achieves $\mathcal{O}(\epsilon^{-2})$ communication complexity in federated min-max learning.
> To our best knowledge, the state-of-the-art results are summarized in Table 1 in our paper and Table 1 in [15], which provide a comprehensive and detailed complexity comparison of existing min-max algorithms.
> We would highly appreciate if the reviewer could provide some references on `the prior work achieved $\mathcal{O}(\epsilon^{-2})$ communication complexity.
> In our revision, we will be more than happy to cite them, improve the literature review section, and provide a more comprehensive comparison.
>
> **Ref:**
>
> [R1] Cutkosky, Ashok, and Francesco Orabona. "Momentum-based variance reduction in non-convex sgd." Advances in neural information processing systems 32 (2019).

---

### Official Review · Reviewer_ACTs · 2022-07-26

**Rating:** 6
**Confidence:** 3
**Soundness:** 4 excellent
**Presentation:** 2 fair
**Contribution:** 3 good

**Summary:**

The authors provide a variance reduction algorithm SAGDA for federated min-max optimization which covers Federated SGDA as a special case. They analyze the theoretical convergence and communication costs of these algorithms improving upon existing rates and matching that of federated convex optimization, obtaining the "linear speedup in the number of workers". Further, they compare these algorithms against existing baselines for federated min-max optimization and obtain better performance empirically.


**Questions:**

- What is the effect of $\kappa$ in the convergence rates and how does it compare to other existing federated min-max algorithms?
- How does SAGDA compare to SCAFFOLD? Is this comparison equivalent to that of comparing GDA methods to GD methods in the centralized setup or does federation change something?
- What are the key ideas in the proof technique which allow for better rates? From what I could understand in the appendix, the authors use the Error-Feedback framework with control variates for the update steps of $x$ and $y$, extending that of SCAFFOLD, but I would like to know if there were additional ideas needed for the proof to go through?


**Limitations:**

Most of the suggested improvements have been mentioned in Strengths and Weaknesses.

**Strengths And Weaknesses:**

### Strengths
- The proposed algorithm SAGDA achieves best possible convergence and communication under standard assumptions, and can also handle arbitrary client heterogeneity
- Using the analysis of SAGDA for Federated SGDA provides better convergence rates over existing analyses.
- The experimental results sufficiently validate the theoretical rates, and SAGDA outperforms existing methods.

### Weaknesses
- Presentation : The presentation of results and their discussion could be improved substantially. Especially for Theorems 1 and 2, most of the math could be simplified and moved to the appendix keeping only $\mathcal{O}(\cdot)$ rates in the main paper. Further, there were several typos in the equations or terms that were not defined( some of them are mentioned in minor issues) . The paper would greatly benefit by improving the presentation.
- Comparison to SCAFFOLD[1] : SAGDA seems to be the extension of SCAFFOLD to min-max optimization. Similar to SCAFFOLD, the analysis technique obtains the best possible rates for the control-variate free version, Federated SGDA. Therefore, I feel it is important to appropriately credit the SCAFFOLD paper. Further, a comparison of SCAFFOLD v/s SAGDA can help establish the difference between variance-reduced methods in convex optimization and min-max optimization.
- Effect of condition number $\kappa$: It would have been interesting to see if SAGDA achieves optimal $\kappa$-dependence in addition to improving the sample and communication complexity.
- Experiments on deep learning tasks: Since min-max optimization is a core component in training GANs, and the performance of variance reduction methods in deep learning is not easy to handle [2], conducting a few toy experiments on deep learning tasks would further improve the paper.

### Minor Issues
- Proof Idea:  Since the theoretical results of this paper improve upon the existing rates, even for Federated SGDA, highlighting the key ideas in the convergence proof which allow this improvement would help understand what previous analyses missed.
- Extension to different classes of non-convex objectives: The authors only consider nonconvex-PL objectives, while the analysis technique might also be applied to more classes, for instance, those in  Table 1 in [3]). It would also be interesting to see if this has the same effect of mitigating heterogeneity and if it improves upon existing rates.
- Typos and issues with presentation:
    - In Section 2 (Related Work), Lines 101-104, it is not clear what $K$ is. Later in the paper, it is defined as the number of local steps, but it would be better to mention this here as well.
    - Eq.(2) $\nabla_x f_i(z_{t,i}^j, \xi_{t,i}^k)$ -> $\nabla_x f_i(z_{t,i}^k, \xi_{t,i}^k)$
    - Unnumbered Eq after Line 171, there is an additional $\xi_{t,i}^k$.

### References
1. Karimireddy, S.P., Kale, S., Mohri, M., Reddi, S., Stich, S. &amp; Suresh, A.T.. (2020). SCAFFOLD: Stochastic Controlled Averaging for Federated Learning. *Proceedings of the 37th International Conference on Machine Learning*, in *Proceedings of Machine Learning Research* 119:5132-5143 Available from https://proceedings.mlr.press/v119/karimireddy20a.html.
2. Defazio, A., & Bottou, L. (2019). On the Ineffectiveness of Variance Reduced Optimization for Deep Learning. In Advances in Neural Information Processing Systems. Curran Associates, Inc..
3.
Sharma, P., Panda, R., Joshi, G. &amp; Varshney, P.. (2022). Federated Minimax Optimization: Improved Convergence Analyses and Algorithms. *Proceedings of the 39th International Conference on Machine Learning*, in *Proceedings of Machine Learning Research* 162:19683-19730 Available from https://proceedings.mlr.press/v162/sharma22c.html.

---

> ### Author Response · Authors · 2022-08-02
> **Response to Reviewer ACTs [1/3]**
>
> Thank you very much for your review and constructive comments, which helped us significantly improve the quality of this paper. In this revision, we have carefully revised our paper based on your comments, questions, and suggestions. The detailed point-by-point responses are as follows:
>
> ### **Weaknesses**
>
> > 1. Presentation : The presentation of results and their discussion could be improved substantially. Especially for Theorems 1 and 2, most of the math could be simplified and moved to the appendix keeping only $\mathcal{O}(\cdot)$ rates in the main paper. Further, there were several typos in the equations or terms that were not defined( some of them are mentioned in minor issues) . The paper would greatly benefit by improving the presentation.
>
> **Our response:** Thanks for your constructive comments. We will simplify the presentation of the theoretical results as you suggested and provide clearer definitions of the parameters. We will also correct the typos.
>
> > 2. Comparison to SCAFFOLD[1] : SAGDA seems to be the extension of SCAFFOLD to min-max optimization. Similar to SCAFFOLD, the analysis technique obtains the best possible rates for the control-variate free version, Federated SGDA. Therefore, I feel it is important to appropriately credit the SCAFFOLD paper. Further, a comparison of SCAFFOLD vs SAGDA can help establish the difference between variance-reduced methods in convex optimization and min-max optimization.
>
> **Our response:** Thanks for your comments and suggestions. Despite different optimization problems (min-max vs. minimization), we agree that our SAGDA algorithm and SCAFFOLD share some technical similarities in utilizing control-variates to achieve better convergence rates. Thus, we have given and would give more credit to SCAFFOLD paper as you suggested. We will add comparisons between SCAFFOLD and SAGDA, and further discussion of variance reduction methods in min-max and minimization problems.
> In short, compared with SCAFFOLD for minimization problems, the main challenge for SAGDA for minimax problems is that the extra error introduced by the parameter $y$. However, it is not a trivial task to analyze the error induced by $y$ due to the maximization operator and PL condition (which is weaker than strongly concave). These challenges motivate us to construct a new potential function and extra analysis effort for the proof.
>
> > 3. Effect of condition number $\kappa$: It would have been interesting to see if SAGDA achieves optimal $\kappa$ dependence in addition to improving the sample and communication complexity.
>
> **Our response:** Thanks for your insightful comments. The $\kappa$ dependence of SAGDA is $\min_{t \in [T]} \| \Phi(x_t) \|^2 = \mathcal{O}(\kappa^2)$, which is the same as that in [3] (see Thm 1 in [R1]). We believe that this order matches the state-of-the-art order but we are not sure whether this $\kappa$-dependence is optimal since there is no known lower bound for the dependence on the condition number $\kappa$.
> To see how our SAGDA algorithm can achieve this $\kappa$-dependence, note that we require the learning rates $\eta_y/\eta_x = \Theta(\kappa^2)$ (see condition of  $a_2$ in the paper ) due to the nonsymmetric nature of min-max problems, which is consistent with that of the centralized GDA (in Ref. [36]). Substituting this result into the convergence bound, we can see that the condition number dependence is $\kappa^2$.
>
> [R1] Sharma, Pranay, et al. "Federated minimax optimization: Improved convergence analyses and algorithms." International Conference on Machine Learning. PMLR, 2022.

---

> > ### Author Response · Authors · 2022-08-02
> > **Response to Reviewer ACTs [2/3]**
> >
> > > 4. Experiments on deep learning tasks: Since min-max optimization is a core component in training GANs, and the performance of variance reduction methods in deep learning is not easy to handle [2], conducting a few toy experiments on deep learning tasks would further improve the paper.
> >
> > **Our response:**
> > Thanks for your suggestions. In our experiment section, we focused on two learning tasks: logistic regression and AUC maximization, which are the benchmark tasks in minimax learning literatures.  We do agree with the reviewer that our paper will benefit from having further experimental results on GANs. In this rebuttal period, we have run additional experiments on GANs. In this experiment, we aim to optimize the following loss function:
> > $$f_{i}(\mathbf{x}, \mathbf{y}) = E_{ a_i \sim P_{true} } [ \log D_{\mathbf{y}}\left( a_i \right)] + E_{z \sim P_{z}} \left[ \log \left(1-D_{\mathbf{y}} \left( G_{ \mathbf{x}} (z) \right) \right) \right]$$
> > where $a$ is the data point and $P_{true}$ is the  distribution of the true samples,  $G_{\mathbf{x}}$ is the generator with parameter $x$ and $D_{\mathbf{y}}$ the discriminator with parameter $\mathbf{y}$.
> > $z$ denotes the input noise vector and $P_{z}$ is the prior distribution of the noise vector for generating samples.
> > We have tested the convergence performance of our algorithms using the MNIST dataset. We summarized the results in Table 1
> >
> > We can observe that both our proposed algorithms FSGDA and SAGDA have better convergence performance compared with the baselines:
> >
> > Table1.$||\nabla\phi(x)||^2$ vs. communication round.
> >
> > |Algorithms$\backslash$Communication round|0|20|40|100|
> > |:---:|:------:|:------:|:------:|:------:|
> > |SAGDA-option1|$7.34\times10^{-2}$|$7.29\times10^{-2}$|$7.25\times10^{-2}$|$7.19\times10^{-2}$|
> > |SAGDA-option2|$7.34\times10^{-2}$|$7.29\times10^{-2}$|$7.25\times10^{-2}$|$7.19\times10^{-2}$|
> > |FSGDA|$7.34\times10^{-2}$|$7.30\times10^{-2}$|$7.26\times10^{-2}$|$ 7.19\times10^{-2}$|
> > |CD-MAGE+|$7.34\times10^{-2}$|$7.31\times10^{-2}$|$7.28\times10^{-2}$|$7.23\times10^{-2}$|
> > |CD-MA|$7.34\times10^{-2}$|$7.32\times10^{-2}$|$7.30\times10^{-2}$|$7.24\times10^{-2}$|
> > |Parallel-SGDA|$7.34\times10^{-2}$|$7.33\times10^{-2}$|$7.32\times10^{-2}$|$7.29\times10^{-2}$|
> >
> > ### **Minor Issues**
> >
> > > 1. Proof Idea: Since the theoretical results of this paper improve upon the existing rates, even for Federated SGDA, highlighting the key ideas in the convergence proof which allow this improvement would help understand what previous analyses missed.
> >
> > **Our response:** Thanks for your suggestions. The key ideas for the proof contain three important components: 1) the use of control variates in the local update can effectively mitigate the data heterogeneity, and thus removing any assumption on the non-i.i.d. datasets; 2) we utilize two-sided learning rates for parameters $x$ and $y$. Specifically, we use a small local learning rate to properly control the model discrepancy among clients (i.e., the ``client drift'' phenomenon) and a relatively large global learning rate for model aggregation; 3) we design a new potential function to cancel out the errors in both $x$- and $y$-variables in local update steps. We appreciate your suggestions and will add the explanations of these key ideas in the revision.
> >
> > > 2. Extension to different classes of non-convex objectives: The authors only consider nonconvex-PL objectives, while the analysis technique might also be applied to more classes, for instance, those in Table 1 in [3]). It would also be interesting to see if this has the same effect of mitigating heterogeneity and if it improves upon existing rates.
> >
> > **Our response:** Thanks for your comments and references.
> > This work is focused on the non-convex-PL objectives, but we believe that our SAGDA algorithm can be directly applied to solve non-convex-strongly-concave objectives and maintain the same convergence rate by using strongly concave condition on $y$ to prove the inequality in Line 479 in the supplementary material.
> > However, for non-convex-concave (NC-C) and non-convex-1-Point-Concave (NC-1PC) objectives, we think our control-variate technique would also be able to handle non-i.i.d. data, but it requires extra effort to obtain the convergence rate.
> > Thanks for your suggestions! We agree that it would be a very interesting future direction to the investigate the performance of our algorithms under the NC-C and NC-1PC objective classes.

---

> > > ### Author Response · Authors · 2022-08-02
> > > **Response to Reviewer ACTs [3/3]**
> > >
> > > > 3. Typos and issues with presentation
> > >
> > > **Our response:** Thanks for pointing them out. We will fix them in the paper.
> > >
> > > ### **Questions**
> > >
> > > > 1. What is the effect of $\kappa$ in the convergence rates and how does it compare to other existing federated min-max algorithms?
> > >
> > > **Our response:** Thanks for your comments. The dependence of $\kappa$ in the convergence rate is $\mathcal{O}(\kappa^2)$, which matches the state-of-the-art results. Please see our response to your Weakness 3.
> > >
> > > > 2. \How does SAGDA compare to SCAFFOLD? Is this comparison equivalent to that of comparing GDA methods to GD methods in the centralized setup or does federation change something?
> > >
> > > **Our response:**  Please see our response to your Weaknesses 2.
> > >
> > > > 3. What are the key ideas in the proof technique which allow for better rates? From what I could understand in the appendix, the authors use the Error-Feedback framework with control variates for the update steps of x and y, extending that of SCAFFOLD, but I would like to know if there were additional ideas needed for the proof to go through?
> > >
> > > **Our response:** Thanks for your comments. The answer is ``Yes," i.e., there are additional ideas for the proof beyond SCAFFOLD. The main ideas contain two parts: 1) control variates; and 2) a new potential function. Similar to SCAFFOLD, we use the control variates for the update steps of $x$- and $y$-variables. By doing so, the distance of local steps in each client can be appropriately bounded with sufficiently small local learning rates, and the error bound is independent of the data heterogeneity; The *new and additional idea* for the proof lies in the the second component, where we construct a new potential function to cancel out the error induced by $y$. The error induced by $y$ in local steps is a new challenge (containing $\nabla_y f(x, y)$). By using our new and carefully designed potential function ($\mathcal{L}$ in the paper), we are able to cancel it out.

---

> > > > ### Comment · Reviewer_ACTs · 2022-08-09
> > > > **Final Comments**
> > > >
> > > > I would like to thank the authors for their detailed rebuttal. Almost all my questions have been answered and I hope that the authors would incorporate these suggestions in the future versions. I have summarized these below.
> > > >
> > > > - Improve presentation of results
> > > > - A small proof sketch to highlight their analysis
> > > > - Comparison to SCAFFOLD
> > > > - Thorough experiments of SAGDA on GANs or other non-convex problems.
> > > >
> > > > Adding these changes would greatly improve the quality of this paper, and as most of these issues were addressed here partially, I am increasing my score for this paper.

---

> > > > > ### Author Response · Authors · 2022-08-09
> > > > > **Response to Reviewer ACTs' Final Comments**
> > > > >
> > > > > Thanks so much for going through our previous response carefully and raising your score! We will definitely incorporate your comments and suggestions in our revision. Thanks!
> > > > >
> > > > > Best,
> > > > > Authors

---

### Meta-Review · Area_Chair_34GX · 2022-08-25

**Recommendation:** Accept
**Confidence:** Certain

**Metareview:**

While the reviewers have shown some concerns regarding presentations, the paper has substantial contribution. It is suggested that the authors improve their presentation taking the suggestions from the reviewers: this paper will gain from clarifying the contributions. Overall the novelty of min-max optimization coupled with interesting results warrants acceptance.

**Award:**

No

---

### Decision · Program_Chairs · 2022-09-14

Accept